# Alternating quarantine for sustainable epidemic mitigation

Dror Meidan [1], Nava Schulmann[2,3], Reuven Cohen [1], Simcha Haber [1], Eyal Yaniv[4], Ronit Sarid[5] &
Baruch Barzel [1,6]✉

Absent pharmaceutical interventions, social distancing, lock-downs and mobility restrictions remain our prime response in the face of epidemic outbreaks. To ease their potentially devastating socioeconomic consequences, we propose here an alternating quarantine strategy: at every instance, half of the population remains under lockdown while the other half continues to be active - maintaining a routine of weekly succession between activity and quarantine. This regime minimizes infectious interactions, as it allows only half of the population to interact for just half of the time. As a result it provides a dramatic reduction in transmission, comparable to that achieved by a population-wide lockdown, despite sustaining socioeconomic continuity at ~50% capacity. The weekly alternations also help address the specific challenge of COVID-19, as their periodicity synchronizes with the natural SARS-CoV-2 disease time-scales, allowing to effectively isolate the majority of infected individuals precisely at the time of their peak infection.

[1] Department of Mathematics, Bar-Ilan University, Ramat-Gan, Israel. [2] Department of Mechanical Engineering, Politecnico di Milano, Milan, Italy. [3] MIMESIS, Inria, Strasbourg, France. [4] Graduate School of Business Administration, Bar-Ilan University, Ramat-Gan, Israel. [5] Faculty of Life Sciences & Institute of Nanotechnology and Advanced Materials, Bar-Ilan University, Ramat-Gan, Israel. [6] Gonda Multidisciplinary Brain Research Center, Bar-Ilan University, Ramat-Gan, Israel. ✉email: baruchbarzel@gmail.com

Battling the spread of SARS-CoV-2, most countries have resorted to social distancing policies, imposing restrictions[1], from complete lockdowns to severe mobility constraints[2–5], gravely impacting socioeconomic stability and growth. Current observations indicate that such policies must be put in place for extended periods (typically months) to avoid the reemergence of the epidemic once lifted[6–8]. This, however, may be unsustainable, as individual social and economic needs will, at some point, surpass the perceived risk of the pandemic[9].

Consequently, in practice, many countries have been experiencing sporadic instances of social restrictions[10–12], from extensive periods of school closure[13] to bans on different social and economical activities[14,15]—often lacking a systemic strategy for containment[16]. The socioeconomic consequences are devastating, from political unrest, to grave economical losses and deteriorating mental health of at-risk demographics[17]. Indeed, pandemics—both current and future—expose a crucial vulnerability of modern society[18,19], calling on us to design socioeconomically sustainable response protocols, in the absence of therapeutic interventions.

We, therefore, examine here an alternating quarantine (AQ) strategy, tailored and tested for our immediate threat of COVID-19, but equally relevant to other pandemic spreading scenarios. The AQ strategy is based on two principles: (i) Complete isolation of symptomatic individuals and their household members[1]; (ii) partitioning of the remaining households into two cohorts that undergo weekly successions of quarantine and routine activity. Other periodic cycles, e.g., bi-weekly, or 5 working days vs. 9 quarantine days, may also be considered. The partition, we emphasize, must be at the household level, guaranteeing all cohabitants are in the same cohort. This way, while Cohort 1 remains active, Cohort 2 stays at home and vice versa, ensuring little interaction between the cohorts (Fig. 1d). This provides highly efficient mitigation, alongside continuous socioeconomic productivity, in which half of the workforce remains active at each point in time.

The AQ strategy limits social mixing[12] while providing an outlet for people to sustain their economic and social routines. Its efficiency is rooted in two independent mitigating effects: (i) dual-partition of population and time, and (ii) synchronization with the disease cycle (Fig. 1). To understand the effect (i), consider the impact of AQ on the transmission rate. First, it splits the population into two isolated cohorts—this reduces the number of

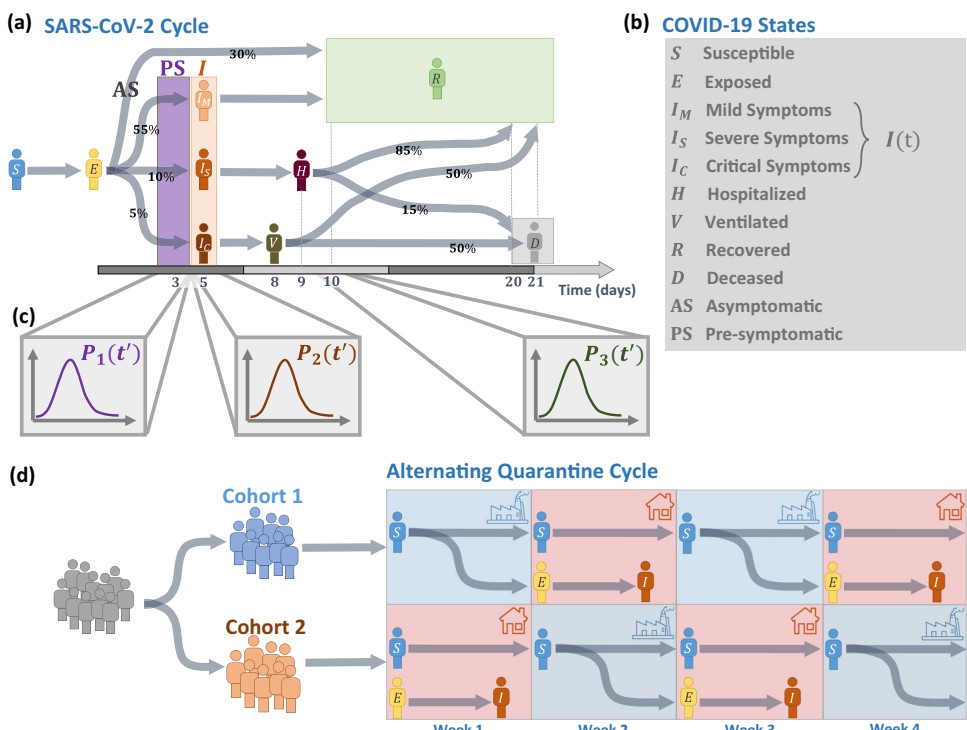

**Fig. 1 The cycles of SARS-CoV-2 and COVID-19 vs. those of the alternating quarantine strategy. a** We collected data on the transitions between the SARS-CoV-2 and COVID-19 states and constructed the characteristic disease cycle. Upon exposure (*E*) individuals enter an average 5 day incubation period prior to developing symptoms—mild (*$I_M$* at a rate of 55%), severe (*$I_S$*, 10%), or critical (*$I_C$*, 5%). The remaining ~30% are asymptomatic (AS). Infectiousness begins typically 3 days after exposure for symptomatic carriers, and 4 days for the asymptomatic (AS). The *infection window* (violet) captures the invisible presymptomatic (PS) spreading phase, in which individuals are infectious, but lack symptoms. Upon the onset of symptoms, infected individuals are isolated and cease to infect others. Consequently, asymptomatic individuals have a longer infection window, which extends until their transition to *R*. As the disease progresses a fraction of the infected population may require hospitalization (*H*) or ventilation (*V*), leading, to some probability of mortality (*D*). **b** The compartments of the COVID-19 cycle. We denote by *I(t)* the unity of all *symptomatic* individuals (*I = $I_M$ + $I_S$ + $I_C$*). This corresponds to the diagnosed case count in each country (Fig. 3), which covers mainly the patients who exhibit symptoms. **c** While the illustrated cycle in (**a**) captures the average transition times between all states, in reality, some level of variability exists across the population. This is captured by the distribution $P_i(t')$. For example, the individual transition time from *E* to PS, whose average is 3 days, is extracted from $P_1(t')$ (purple). **d** Alternating quarantine (AQ) splits the population into separate cohorts that alternate between periods of activity (going to work, blue) and inactivity (staying at home, red). Following their active week, (week 1) individuals in Cohort 1 may become exposed (yellow), in which case they will sit out their suspected pre-symptomatic period at home (week 2). By the end of their quarantine week, they will likely develop symptoms (orange) and remain in isolation until their full recovery. Those who did not develop symptoms during their week of quarantine are most likely uninfected (blue) and can resume activity in their upcoming active shift (week 3). Therefore the AQ cycle behaves as an effective ratchet, consistently quarantining the invisible spreaders, and hence, removing, with each weekly succession, infectious individuals from the active population.

infectious encounters, as, indeed, classrooms, offices, and public places operate at half their usual density. Therefore, individuals interact with only half their usual contacts. On top of that, each cohort is only active for half of the time, one week out of two. This further attenuates infections within each cohort by, roughly, an additional factor of one-half.

This dual-partition effect, relevant for any general pandemic, is reinforced by an additional factor, unique to COVID-19—the synchronization of AQ with the natural disease cycle, i.e., factor (ii) above. This treats one of the main obstacles for COVID-19 mitigation—the fact that while we isolate the symptomatic patients, exposed individuals become infectious a few days prior to the onset of symptoms[11,20–24] (Fig. 1a). During this pre-symptomatic stage, they behave as invisible spreaders, who continue to interact with their network, unaware of their potential infectiousness[25–32]. To illustrate AQ's remedy, consider an individual in Cohort 1 who was active during week 1, and therefore might have been infected. This individual will soon enter their presymptomatic stage, precisely the stage in which they are invisible, and hence contribute most to the spread. However, according to the AQ routine, they will be confined to their homes during week 2, and consequently, they will be isolated precisely during their suspected presymptomatic period. If, by the end of week 2 they continue to show no symptoms, most chances are that they are, in fact, healthy, and can, therefore, resume activity in week 3 according to the planned routine. Conversely, if they do develop symptoms during their quarantine, they (and their cohabitants) must remain in isolation, similar to all symptomatic individuals. Hence, the weekly succession is in resonance with the natural SARS-CoV-2 disease cycle[10], and in practice, leads to isolation of the majority of invisible spreaders. If implemented fully, it guarantees, in each bi-weekly cycle, to prune out the infectious individuals and sustain an active workforce comprising a predominantly uninfected population.

Using COVID-19 as our test case, we examine below the performance of AQ and discuss practical aspects pertaining to its implementation, from guidelines on how to partition the social network to the treatment of limited social compliance.

## Results

**Modeling the spread of SARS-CoV-2**. *Social network*: We constructed a population of $N \sim 10^4$ individuals, comprising $M = 4 \times 10^3$ separate households, and tracked their sequence of social interactions at 15 min resolution over the course of $T = 150$ days. These interactions are governed by two separate networks: daytime interactions at work, school, or other public places are driven by the external network $A_{ij}$. This represents an $N \times N$ network with degree distribution $P(k)$, designed to capture out-of-home social activity (Fig. 2a, orange). In-house interactions, taking place predominantly during the after-hours, are governed by $B_{ij}$, a network of $M$ isolated cliques, capturing households (Fig. 2a, blue). The size of these cliques $m$ is extracted from the empirically obtained[33] household size distribution $P(m)$.

To capture the temporal nature of the interactions, each link in $A_{ij}$ and $B_{ij}$ switches between periods of activity, i.e., collocation of $i$ and $j$, vs. intermittent periods, in which the link remains idle. Infection between $i$ and $j$ may occur during the instances in which the $i$, $j$ link is active. These instances of activity/inactivity, extracted at random, represent the sporadic nature of human interaction and allow us to track the viral spread under highly realistic conditions. To design a typical daily cycle, the external links $A_{ij}$ are predominantly active during the day (Fig. 2c), while in-house interactions occur primarily at the evening/night-time hours (Fig. 2d). During quarantine, such as under the AQ routine,

or if a household member $i$ exhibits symptoms, the relevant $A_{ij}$ links remain idle, while $B_{ij}$ becomes activated also during the day.

Each of the networks $A_{ij}$ and $B_{ij}$ is characterized by two independent parameters (Fig. 2b). The first captures the probability of link-activation at each 15-min interval, determining the links' mean daily duration of the activity. We denote this duration by $T_1$ for $A_{ij}$ and $T_2$ for $B_{ij}$. Realistically we expect $T_1 < T_2$, capturing the fact that cohabitants, when at home, spend more time in potentially infectious interactions than, e.g., office-mates during work hours. The second parameter is the probability of transmission per interaction, set to $p_1$ and $p_2$ for $A_{ij}$ and $B_{ij}$, respectively. Also here we assume that typically $p_1 < p_2$, as in-house interactions, often between family members, are potentially more infectious than the social interactions of $A_{ij}$. For example, parents tending to children or siblings interacting physically, are more likely to lead to infection, than co-workers sharing a meeting room.

*Disease cycle*: In Fig. 1a, we present the SARS-CoV-2 characteristic infection cycle. Upon exposure ($E$) individuals enter a presymptomatic period, which lasts, on average ~5 days, after which they begin to exhibit mild ($I_M$), severe ($I_S$), or critical ($I_C$) symptoms, leading to hospitalization ($H$), and in certain cases also to ventilation ($V$). Approximately 2 days prior to the onset of symptoms the exposed individuals become infectious, hence, on average, the infectious phase begins 3 days after initial exposure[20]. Spreading the virus continues until the onset of symptoms, at which point the infected individuals, together with their cohabitants, enter isolation and cease to contribute to the external spread ($A_{ij}$). Of course, in-house transmission ($B_{ij}$) continues also during home-isolation. A fraction of the exposed individuals remain asymptomatic (AS), and hence do not isolate, throughout their entire infectious period, beginning on average 4 days posterior to exposure[34]. Hence, the symptomatic carriers spread the disease within an average window of ~2 days (purple), while the AS carriers continue to infect others until their immune response clears the virus.

These time-scales represent the average infection cycle, which, in reality, may exhibit variability across the population[11,22–24,34–37]. This is especially relevant regarding the time for the appearance of symptoms, which, if delayed beyond 1–2 weeks, may lead to an infectious crossover between successive terms of activity, e.g., if a person is infected in week 1, and then, lacking symptoms, resumes activity in week 3 (see Fig. 1d). Therefore, for each of the relevant time-scales, e.g., the time from exposure to infectiousness, or the time to develop symptoms, we consider not just the average, but the complete distribution across the population (Fig. 1c). For example, the probability density function $P_1(t')$ captures the fraction of exposed individuals who exhibits symptoms within $t \in (t', t' + \mathrm{d}t')$ days from exposure. Similarly, $P_2(t')$ characterizes the transition times between exposure and AS infectiousness. The broader are $P_i(t')$, the greater is the individual variability in transition times between the different disease states. Here we extract $P_i(t')$ from a Weibull distribution, as indicated for SARS-CoV-2[23,24,37], and observed also for other infections of Corona type viruses[34]; see Supplementary Notes 1.3 and 4.1.

**Characterizing the spread**. Taken together, our modeling framework is designed to capture the spread of SARS-CoV-2 in a detailed fashion, therefore, characterized by an array of relevant parameters. The majority of these parameters can be extracted from empirical data. For example, the transition rates of Fig. 1a's disease cycle, or the household size distribution $P(m)$, helping us structure $B_{ij}$—are all empirically accessible. The remaining parameters $P(k)$, $T_1$, $T_2$, $p_1$, $p_2$, however, are unknown. Hence, we

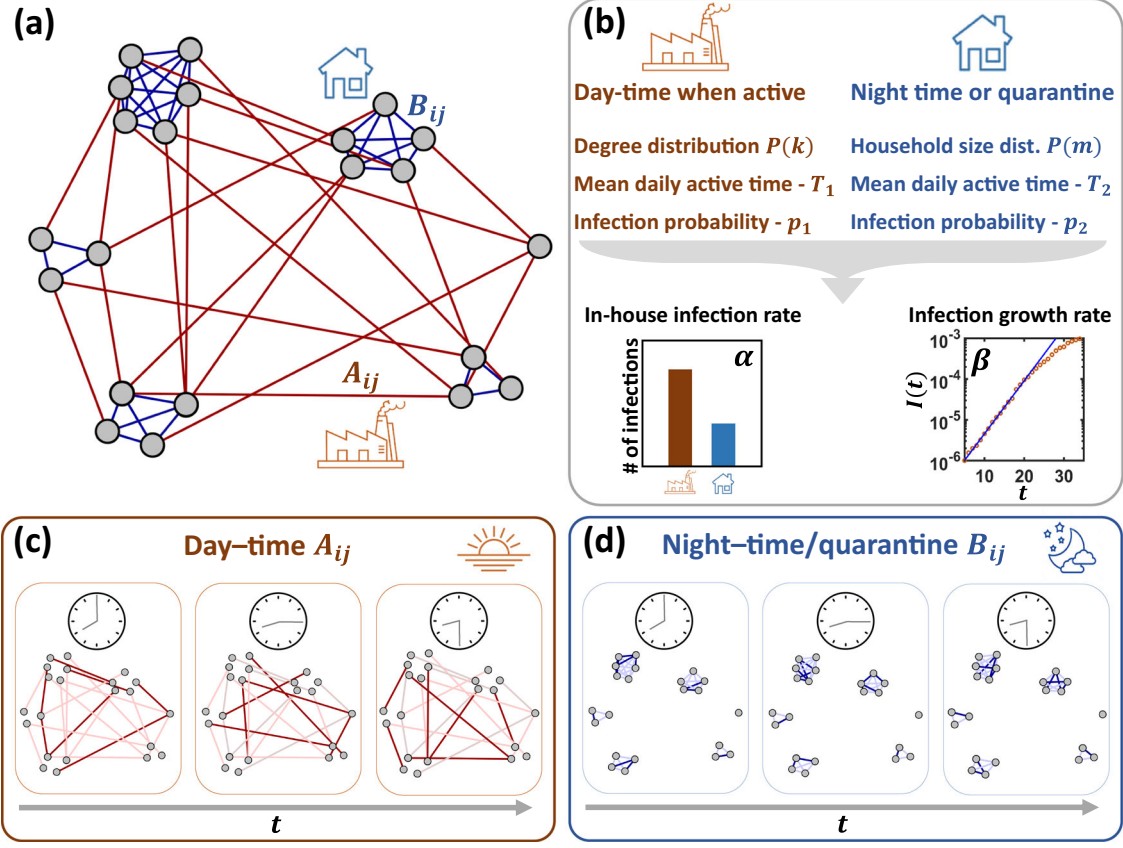

**Fig. 2 Modeling the spread of COVID-19. a** We constructed a social network with two types of links: $A_{ij}$ (red) are external links, representing out-of-home interactions at work, public places, and social gatherings; $B_{ij}$ (blue) capture cohabitant links, capturing separate households. In our setting the network includes $N = 10^4$ individuals split among $M = 4 \times 10^3$ households. **b** The links in $A_{ij}$ are active primarily during the day-time, and only between individuals who are not under quarantine. The in-house connections $B_{ij}$ are activated in the after-hours, or all day when under quarantine. Both networks are characterized by the degree/household size distributions $P(k)$ and $P(m)$. Their links exhibit sporadic instances of activity, capturing times when $i$ and $j$ are collocated, and therefore engage in potentially infectious interaction. Hence, they are characterized by two parameters: $T_1$, $T_2$ capture the typical time per day in which the links are active, and $p_1$, $p_2$ capture the probability of infection at each such instance of the activity. This captures the fact that (i) cohabitant links are, typically, more active than social links ($T_2 > T_1$); (ii) when activated, cohabitant interactions are often more intimate and therefore potentially more infectious ($p_2 > p_1$). Together these six parameters, balancing the relative roles of $A_{ij}$ and $B_{ij}$ in the virus transmission, give rise to two observable parameters that help characterize the contagion: $\alpha$ in (2), quantifying the contribution of external (red) vs. in-house (blue) transmission to the spread; $\beta$ in (1), describing the proliferation rate of the virus. **c** We simulated social activity over a period of $T = 150$ days, at 15-min resolution. At each 15-min instance, a fraction of the links is active (dark red), while the other remains idle. Transmission between $i$ and $j$ can only occur (with probability $p_1$) at times when $A_{ij}$ is active (on average $T_1$ percent of the time). External links $A_{ij}$ are active primarily during the day, excluding periods of quarantine. **d** $B_{ij}$ undergoes a similar pattern of activity/idling, with parameters $T_2$, $p_2$, primarily in the evening/night-time, or all day during the quarantine.

examine different spreading scenarios to test our strategy's sensitivity to discrepancies in these parameters. For instance, we consider both a random $A_{ij}$, for which $P(k)$ is bounded and a scale-free $A_{ij}$, where $P(k)$ is fat-tailed (Supplementary Note 2). Similarly, we assign different values to $T_1$ and $T_2$ to examine the variable balance between in-house and external transmission.

Once these unknown parameters are set, they help characterize the spread along two dimensions (Fig. 2b):

- The growth rate $\beta$, quantifies the level of spread by tracking the initial exponential proliferation of infections

$$I(t) \sim e^{\beta t}, \qquad (1)$$

observed at the early stages of the outbreak. Here, $I(t) = I_M(t) + I_S(t) + I_C(t)$ is the number of symptomatic infected individuals. The greater is $\beta$ the more rapid is the spread, hence $\beta$ increases with $T_1$, $T_2$, $p_1$, and $p_2$, all of which contribute to the infectiousness of the disease. The density of the network, i.e. its average external degree $\langle k \rangle$ and household size $\langle m \rangle$ also

both positively contribute to $\beta$, as they allow more potential infectious interactions.

- The in-house infection rate $\alpha$, captures the balance between the contribution of $B_{ij}$ vs. $A_{ij}$ to the spread. To quantify this balance we track all instances of transmission $\theta_{\text{Tot}}$, and extract $\theta_{\text{In}}$, which counts only the cases transmitted via $B_{ij}$ links, i.e. in-house. We then measure the in-house infection rate as

$$\alpha = \frac{\theta_{\text{In}}}{\theta_{\text{Tot}}} \qquad (2)$$

namely the fraction of transmissions that occurred at home.

Similarly to $\beta$, the parameter $\alpha$ is also dictated by the network parameters. A large $T_2$ and $p_2$ will favor in-house transmissions, contributing to $\alpha$, whereas increasing $T_1$, $p_1$ will strengthen the role of external transmissions. The network structure also factors in through the average number of external links $\langle k \rangle$, decreasing $\alpha$, vs. the typical household size $\langle m \rangle$, which increases it. This parameter is especially meaningful in the context of quarantine-

based strategies, which, by design, suppress only external interactions, and therefore become less effective when $\theta_{In}$ is large. Indeed, no matter how strict the quarantine is, it cannot prevent the secondary transmission between household members encapsulated within $\theta_{In}$. In fact, it often increases these in-house infections, as it forces cohabitants to remain close for extended periods. Consequently, a large $\alpha$ can potentially hinder the effectiveness of quarantine in general, and of AQ in particular.

To summarize, in our modeling we vary the *implicit* model parameters $T_1$, $p_1$, $T_2$, $p_2$, $P(k)$, to scan an array of potentially relevant scenarios. Once setting these parameters, we use them to extract two observable parameters $\alpha$ and $\beta$, that directly characterize the nature of the contagion. The severity of the spread is quantified by $\beta$, and the role of the in-house transmission is captured by $\alpha$. The mapping between the model parameters and the observable $\alpha, \beta$ is explained in Supplementary Note 1.4.

**Projecting the spread of SARS-CoV-2.** To evaluate $\beta$ for the unmitigated COVID-19 spread we collected data on the evolution of the epidemic in 12 different countries[38], and examined $I(t)$ at the early stages of the spread, prior to the implementation of social distancing policies (Fig. 3a–l). Fitting to an exponential growth of the form (1) we evaluate $\beta$ in each country, obtaining a mean growth rate of $\beta = 0.26$ days$^{-1}$, an estimate congruent with other independent evaluations[34,35]. To further ground this estimate, we use $\beta$ to evaluate the epidemic reproduction number[39], providing $R_0 \approx 2.4$, again, in line with current valuations (Supplementary Note 1.5). Such fitting, based on the early spreading dynamics, may, generally risk overestimating $\beta$ and $R_0$[40–42].

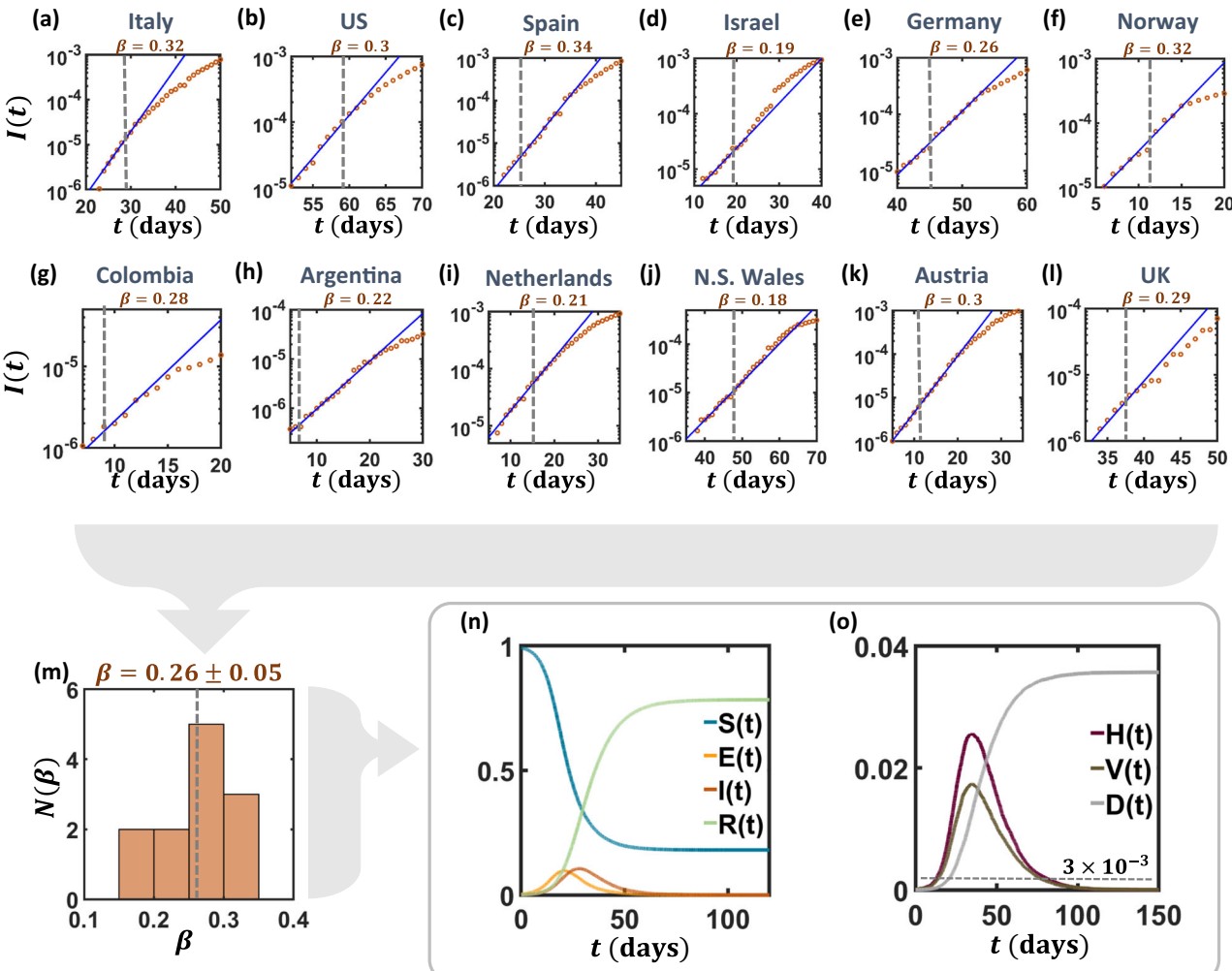

**Fig. 3 Extracting the SARS-CoV-2 infection rate. a–l** We collected data[38] on the infection levels $I(t)$ vs. $t$ (orange circles) in 12 different countries. Fitting $I(t)$ to an exponential of the form (1) we evaluate $\beta$ in each of these countries (blue solid lines). Such exponential growth typically continues for a period of several days posterior to the instigation of a mitigation policy (gray dashed lines). We, therefore, used only the data up to 3 days after the implementation of social distancing to evaluate $\beta$ (Supplementary Note 4.2). **m** Histogram of inferred $\beta$ values across the 12 countries. Infection rates are distributed around an average of $\beta = 0.26$ days$^{-1}$. Hence, in our simulations, we tune the parameters to obtain this growth rate under the absence of all prophylactic measures. In reality, standard behavioral practices, such as personal hygiene or avoidance of physical contact, may push $\beta$ to lower values. To capture this, in our simulations we incorporate three scenarios: worst-case—$\beta \approx 0.26$, intermediate case—$\beta \approx 0.2$ and best case—$\beta \approx 0.15$. **n** Taking $\beta \approx 0.26$ we simulated the projected evolution of the COVID-19 pandemic *alà* Figs. 1 and 2, without any preventive measures. **o** We focus on three crucial parameters that characterize the severity of the projected spread: mortality $D(t)$ (gray), hospitalization level $H(t)$ (purple), and the number of individuals requiring ventilation $V(t)$ (brown). Absent any intervention, $H(t)$ exceeds, by a large margin, the average national hospitalization capacity (dashed gray line), estimated at $3 \times 10^{-3}$ [43]. Results represent an average of over 20 stochastic realizations on a network comprising 4000 households (~$10^4$ individuals).

However, in our analysis below we consider a spectrum of different scenarios, with different $\beta$ values, and hence our results are insensitive to this risk factor.

Setting, as a baseline, $\alpha = 0$ (to be changed below) we can now use our estimated $\beta$ to obtain a projection of the expected evolution of the epidemic (Fig. 3n). We also track the expected fraction of hospitalized ($H(t)$) and ventilated ($V(t)$) individuals (Fig. 3o), which, we find, exceed, by a significant margin, the average national hospitalization capabilities[43]. The expected

mortality is captured by $D(t \rightarrow \infty)$ reaching, absent any mitigation efforts, a level of ~3% (gray). Next, we examine the behavior of COVID-19 under AQ, together with other relevant strategies.

**Mitigating the spread.** To examine the impact of our proposed strategy we track the evolution of $I(t) = I_M(t) + I_S(t) + I_C(t)$. First, we allow the disease to spread unmitigated (Fig. 4a, orange, UM), then at time $t = t_0$ (Supplementary Note 1.2) we instigate our

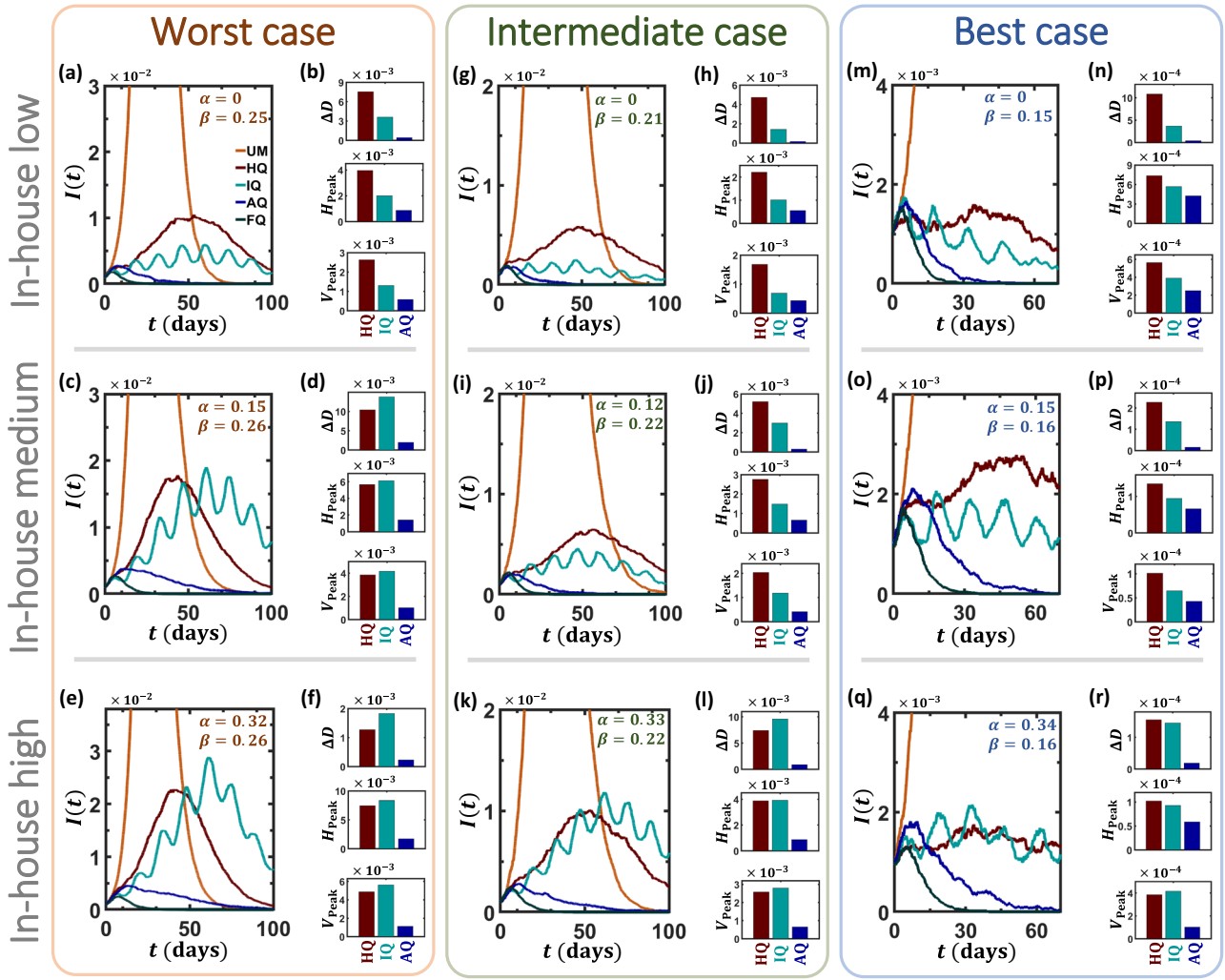

**Fig. 4 The impact of alternating quarantine. a** The infection $I(t)$ vs. $t$ of the unmitigated epidemic (UM, orange), as obtained under $\beta = 0.25$ and $\alpha = 0$. This represents a worst-case scenario, where $\beta$ is taken from the unmitigated spreading data, i.e., lacking prophylactic practices. At $t = t_0$ we instigate four competing mitigation strategies: Full quarantine (FQ, gray), Alternating quarantine (AQ, blue), Intermittent quarantines (IQ, turquoise), and Half quarantine (HQ, red). We find that, barring the idealized FQ, AQ provides the most efficient mitigation, outperforming IQ, and HQ. **b** We used three performance measures to rate mitigation efficiency: Residual mortality $\Delta D$ (3), peak hospitalization $H_{Peak}$ (4), and peak ventilation $V_{Peak}$. In all three indicators, AQ provides the best outcomes by a significant margin, as compared to IQ and HQ. **c–f** To examine the role of in-house transmission, we further tested all strategies under medium ($\alpha = 0.15$) and high ($\alpha = 0.32$) levels of household infections. **g–l** We repeated the same experiment, this time under a ~20% lower $\beta$, capturing the potential effect of complementary prophylactic measures, such as mask-wearing or hygienic behavior. **m–r** In our best-case scenario $\beta$ is further reduced by an additional ~20% factor, this time to $\beta \approx 0.15$. Together, our analysis scans the space of infection ($\beta$) and in-house transmission ($\alpha$) rates, covering a range of potentially relevant conditions for COVID-19 mitigation. We find that, under all conditions, AQ consistently outperforms all contending strategies, providing mitigation that is closest to FQ. Results represent an average of over 20 stochastic realizations on a network comprising 4000 households (~$10^4$ individuals). Mitigation is initiated at time $t_0$, set to be the time when $I(t)$ exceeds $10^{-3}$ (Supplementary Note 1.2). In our simulations, the external network $A_{ij}$ was taken to be an Erdős–Rényi random graph with average degree $\langle k \rangle = 15$; in Supplementary Note 2 we also examine the case of a scale-free $A_{ij}$, obtaining similar results. Note that $\alpha$ and $\beta$ are controlled indirectly, first through the model parameters ($P(k)$, $P(m)$, $T_1$, $T_2$, $p_1$, $p_2$) as illustrated in Fig. 2b, then extracted from the observed stochastic simulation results via Eqs. (1) and (2). Consequently, we cannot control with accuracy the precise values of these parameters. We, therefore, observe slight discrepancies between the different panels, e.g., $\beta = 0.25$ in panel (**a**) vs. 0.26 in panels (**c**) and (**e**).

response. Examining four relevant mitigation strategies, we establish a basis upon which to evaluate AQ's performance.

*Full quarantine—FQ.* This represents the theoretically ideal response, in which all out of home activity is ceased (Fig. 4a–f, gray). The external links $A_{ij}$ become inactive and only in-house transmission ($B_{ij}$) remains, until these secondary infections are also exhausted and the spread reaches a halt. To capture the effect of this in-house perpetuation of the disease we consider several scenarios, from vanishing in-house transmission (Fig. 4a, $\alpha = 0$) to extreme levels of household infections (Fig. 4e, $\alpha = 0.32$). This range of $\alpha$ values corresponds to a 20–40% infection probability between cohabitants[44]. As expected, absent in-house transmission, FQ eradicates the disease extremely efficiently, within ~3 weeks. As $\alpha$ is increased, FQ, expectantly, shows a slight decline in efficiency[8], yet it is still highly effective. Of course, such perfect air-tight quarantine is unrealistic, however, it is useful in the present context, as it provides a baseline for comparison, indeed, setting the bounds for perfect mitigation.

*Alternating quarantine—AQ.* We now examine the AQ strategy (Fig. 4a–f, blue). At $t = t_0$ we partition the households into two equal groups, Cohorts 1 and 2, and have them alternating successively in a bi-weekly cycle of quarantine, in which only $B_{ij}$ is active, vs. regular socio-economic activity, where both $A_{ij}$ and $B_{ij}$ are active. We find, again, that $I(t)$ decays exponentially, albeit at a slower rate, as compared to the perfect FQ. The crucial point, however, is that this decay is now observed, despite the fact that 50% of the population remains continuously active.

For comparison, we consider two natural alternatives to AQ, both designed to sustain socioeconomic activity at a 50% rate:

*Intermittent quarantines—IQ.* In this strategy[45] society as a whole enters a periodic cycle of active vs. quarantined phases, namely the entire population transitions in unison between staying at home and going to work (Fig. 4a–f, turquoise). Originally proposed in the format of a 4:10 periodicity, i.e., 4 days of activity separated by 10 days of quarantine, here we examine its performance under a 7:7 cycle, to be congruent with our implementation of AQ. We find that IQ is significantly less effective than AQ, leading not only to higher peak infection but also to a substantially longer time to return to normalcy.

*Half quarantine—HQ.* Another mitigation alternative that allows a 50% active workforce is based on a selective quarantine, in which only 50% of the population partakes in socio-economic activities, while the remaining half is instructed to stay at home (Fig. 4a–f, red). HQ suppresses the rate of infection by reducing social interactions, i.e., $A_{ij}$, by a factor of roughly one-half. Our simulation results indicate, however, that, similarly to IQ, this reduction is insufficient. Indeed, $I(t)$ continues to proliferate significantly beyond manageable levels, once again, failing to mitigate the disease.

While the majority of infected individuals exhibit mild or no symptoms, a certain percentage may experience severe complications, leading to hospitalization or ventilation, and in some cases to mortality (Fig. 1a). Our mitigation strategy focuses on these undesired paths within the infection track - namely, we aim to decrease mortality $D(t)$ and ensure that at their peak, $H(t)$ and $V(t)$ do not exceed the national hospitalization and ventilation capabilities. To test this we measured the residual mortality

$$\Delta D = D_S(t \to \infty) - D_{FQ}(t \to \infty), \tag{3}$$

where $D_S(t \to \infty)$ is the long term mortality under strategy S, e.g., IQ or AQ, and $D_{FQ}(t \to \infty)$ is the expected mortality under FQ. Indeed, within the framework of quarantine-based strategies,

$D_{FQ}(t \to \infty)$ represents inevitable deaths, rooted in infections that occurred prior to our response, and hence $\Delta D$ captures the additional mortality, that our mitigation failed to prevent. In Fig. 4b, d, f we measure $\Delta D$ under IQ (turquoise), HQ (red), and AQ (blue). The AQ advantage is clearly visible, saving significantly more lives than the competing strategies.

To examine the impact of each strategy on the severe and critical patients, we measure

$$H_{\text{Peak}} = \max_{t=t_0}^{\infty} H(t), \tag{4}$$

capturing the peak hospital occupancy after instigating our response. While IQ and HQ fail to bring $H_{\text{Peak}}$ within capacity ($\sim 3 \times 10^{-3}$), AQ succeeds in sustaining a leveled occupancy. Similar results are also obtained for $V_{\text{Peak}} = \max_{t=t_0}^{\infty} V(t)$.

Taken together, we find that AQ provides the most efficient mitigation, bringing us closest to the ideal performance of FQ, without fully shutting down the economy. To understand the origins of the observed AQ advantage, we first consider its alternatives, IQ, and HQ. The common root of both strategies is that they reduce the level of interaction by a factor of one half. IQ achieves this by decreasing the interaction duration; HQ accomplishes this by diluting the interacting population. In this sense, the strength of AQ is that it benefits from both factors (Fig. 5): partitioning the population into cohorts ensures that only

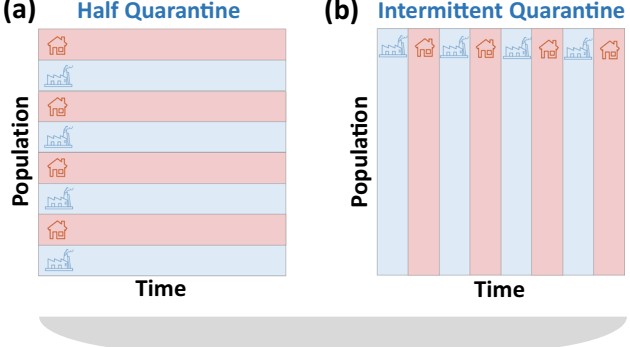

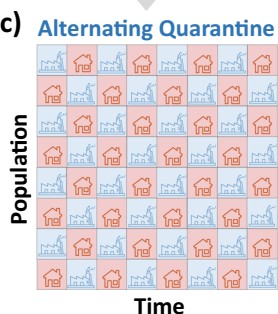

**Fig. 5 The multiplicative effect of alternating quarantine.** We consider three strategies—all allowing socioeconomic activity (blue) vs. quarantine (red) at half capacity. **a** The Half quarantine strategy reduces infection by diluting the active population, hence decreasing the rate of infectious interactions. **b** Intermittent quarantines achieve a similar outcome by diminishing the duration of the activity, hence reducing the time of infectious interactions. **c** Alternating quarantine combines both effects: on the one hand interactions are limited to individuals within each cohort—diluting the population. On the other hand, these cohorts experience intermittent cycles of work/home—diminishing interaction duration. The result is a ~fourfold reduction in transmission, alongside a mere 50% reduction in socioeconomic activity.

half are active at all times—similar to HQ. Yet, the weekly alternations ensure that each cohort remains active only half the time—similar to IQ. This dual partition further reduces infectious interactions without increasing the socioeconomic toll. On top of that, AQ is also tailored specifically to the COVID-19 time-scales, with its weekly periodicity, roughly in-phase with the natural ~5-day cycle of incubation and presymptomatic infection (Fig. 1d). The result is an effective force multiplier, allowing the same amount of net activity—50%—but with a dramatically enforced mitigation effect.

**Synergistic measures**. Our analysis, up to this point, assumed a worst-case scenario, in which, aside from our mitigation strategy (AQ, IQ, or HQ), all other disease parameters remain unchanged. In reality, however, in addition to AQ, or any other strategy for that matter, we can expect, at the least, that standard prophylactic behaviors will continue to be practiced. Indeed, personal hygiene, face-masks, and contact avoidance can reduce infections significantly, without taking any toll on the economy. Therefore, in practice, the infection rate $\beta = 0.26$, inferred in Fig. 3 from the early, pre-mitigation stages of the epidemic, will likely be reduced as we gradually adapt to a prophylactic routine. We, therefore, examine the performance of the different mitigation strategies also under a reduced $\beta$, capturing the synergistic effect offered by such practices. In the intermediate case, we set $\beta \approx 0.21$, a ~20% reduction in the rate of infection (Fig. 4g–l), and as our best-case scenario, we examined $\beta \approx 0.15$, capturing a ~40% drop in infectiousness (Fig. 4m–r). Under these more favorable conditions, AQ's performance approaches even closer to the ideal FQ (e.g., Fig. 4g or m), providing a dramatic reduction in mortality and hospitalization.

More generally, our AQ strategy can, and should, be reinforced by other complementary policies, to ensure mitigation success, from avoiding social gatherings to establishing isolation facilities, with the purpose of reducing in-house transmission. As a specifically relevant example, we consider, in Supplementary Note 3.1, the selective protection of vulnerable populations, addressing a crucial aspect of COVID-19, whose impact on the elderly or on individuals with co-morbidities, is disproportionately more severe[46–48]. In Supplementary Note 3.2, we examine how population-wide testing can further enhance AQ's performance.

All of these policies can be instigated alongside, rather than instead, of AQ. One may also consider alternative periodic cycles[45]. For instance, a 5:9 cycle, in which the active shifts last only 5 days. In this version of AQ, society enters a routine in which each cohort is allowed a 5-day work-week, then observes population-wide quarantine (PWQ) over the weekend. Such adaptations will further improve the performance of AQ beyond its already established effectiveness.

**Alternating vs. PWQ**. The proven advantages of AQ indicate that it is not merely an alternative to IQ or HQ, both partial quarantine strategies, but may actually be confronted against a PWQ. For example, a PWQ at rate $\eta$ requires an $\eta$ fraction of the population to continuously practice quarantine, hence in HQ we have $\eta = 50\%$ and in FQ we have $\eta = 100\%$.

Intuitively, one would expect a PWQ with $\eta > 50\%$ to be more effective than AQ, both in terms of mitigation - isolating larger parts of the population, as well as in terms of implementation - not having to resolve between the two cohorts. Our analysis, however, indicates that AQ has crucial advantages on both fronts. The implementation challenge of PWQ is that it requires people to stay at home for a period of several weeks, in order for the mitigation to take effect. For example, in Fig. 4a we found that a

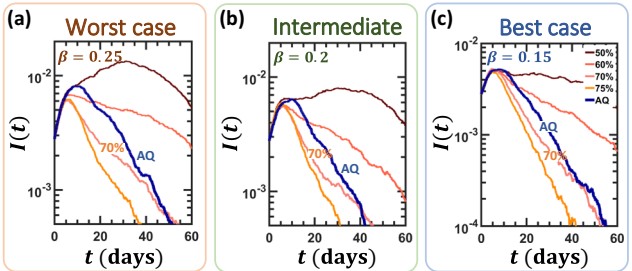

**Fig. 6 Alternating quarantine vs. population-wide quarantine. a** Infection level $I(t)$ vs. $t$ as obtained for $\beta = 0.25$ under Alternating quarantine (AQ, blue). We also examined population-wide quarantines at $\eta = 50, 60, 70,$ and 75% levels (red to yellow). Despite having half of the population active at all times, AQ's mitigation is comparable to that of a 70% population-wide quarantine. Hence, instead of an extremely hurtful socioeconomic shutdown of 70%, nearing the practical upper bound of social distancing policies, AQ offers a similar outcome under a significantly reduced socioeconomic price-tag. **b, c** Similar results are observed also under our intermediate ($\beta = 0.2$) and best case ($\beta = 0.15$) scenarios. Simulations represent an average of over 20 stochastic realizations on a network comprising 4000 households (~$10^4$ individuals). The in-house infection rate was set to the medium level $\alpha \approx 0.15$. Here, $A_{ij}$ is an Erdős–Rényi random graph with $\langle k \rangle = 15$, similar results under a scale-free $A_{ij}$ appear in Supplementary Note 2.

perfectly implemented quarantine $\eta = 100\%$ (FQ), which is, indeed, a theoretical construction only, still required several weeks to achieve a significant gain over the disease[8]. Under these conditions, one cannot implement a truly complete lockdown. Essential services, supply chains, and some parts of the market must remain active since households cannot retain supplies and remain self-sufficient for such extended periods. Therefore, a practical PWQ can at most be implemented at a level of $\eta = 70$–75%[49].

In contrast, the AQ scheme requires individuals to isolate themselves only for a single week at a time. Hence, the quarantined cohort can truly enter, for just 1 week, a complete lockdown regime, in which they avoid purchasing supplies or any other services. Consequently, under AQ, while a larger part of the population is active at all times, the quarantined cohort, can sustain a much stricter lockdown routine. As a result not only is the economy more productive, with 50% of the population continuously active, but the mitigation outcome is also comparable, and under some conditions even superior. To demonstrate this, in Fig. 6 we examine the impact of PWQ, imposed at a level of $\eta = 50, 60, 70,$ and 75% (red to yellow). AQ, we find, is roughly the equivalent of a 70% lockdown (blue). Note that $\eta = 70$–75% represents the practical upper bound for any realistic PWQ. Yet whereas PWQ at such levels severely compromises the economy and imposes significant social and psychological stress, AQ accomplishes a similar effect, while sustaining a productive economy, and allowing a manageable routine for the individual.

## Implementation
*Partitioning*. The AQ strategy works best when the two cohorts are fully separated, lacking all forms of cross-group infection. The partition should, therefore, be implemented at a household level, ensuring all cohabitants are in the same activity/quarantine cycle. A simple way to achieve this is to base the partition on a person's living address. This provides an additional benefit, in the case of apartment buildings, as neighbors, who risk cross-infection through shared building facilities, are included in the same

cohort. Each individual/household will be informed by their local authority of their quarantine schedule, and in parallel, employers will be instructed to resume their activity in shifts, with only half the workforce at a time. Businesses will be held legally liable and incur fines in case of violation.

Instances of conflict between a person's assigned shift and their personal/employer's specific requirements will be resolved on a case by case basis - all while strictly adhering to the household-based partition. To encourage cooperation, and to ensure AQ's smooth implementation, it is best to be as flexible as possible in responding to all individual requests. The resulting cohorts, after accommodating such requests, will likely deviate from an exact balanced cut, however, the crucial point is, that the partition need not be perfect, as, indeed, the cohorts must be decoupled, but not necessarily equal in size. Therefore, there is much room to flexibly address specific constraints or special needs, indeed, easing the psychological and socioeconomic stress as much as possible. In the Methods section, we outline a smooth partitioning scheme, applicable for regional or nation-wide implementation.

*Social compliance.* To engage the population towards cooperation[50], the first step is to communicate the rationale behind AQ, its potential effectiveness, and the individual compliance required for its rapid success. This appeals to people's intrinsic motivation[51], a crucial component of conformity, but often also insufficient due to the tragedy of the commons. We, therefore, list the drivers, that enhance people's desire to cooperate, vs. the inhibitors, that stand in their way[52,53], and set appropriate moderators to enforce the drivers and suppress the inhibitors • Inhibitors (Fig. 7a, b). During its lockdown cycle, the quarantined cohort is required to stay at home for one week, indeed, a challenge, however, being limited in time, it is significantly less stressful than extended several week quarantines. We identify four motivators to violate the quarantine: Business—going to work, schooling—arrangements for child care, services and supplies—visiting public market places or service centers, and outdoor activities—exercise or strolling with children or pets. Of these, the latter, being in the open, is least risky, and also practically unavoidable, as young children and pets require routine outdoor activity. We, therefore, focus on moderators especially for the first three inhibitors • Moderators (Fig. 7c). While cooperation can be achieved via coercion, e.g., law enforcement, it is most effectively garnered by creating supporting frameworks. For example, in the AQ framework, defection for business and schools is simply not possible. Indeed, since businesses are legally required to divide their workforce into shifts, one cannot go to work out of the cycle. Similarly, schools will not admit children who are not in the presently active cohort. Therefore, the main challenge is to deter violators from visiting public places for supplies or services. This can be achieved by (i) instructing the population to prepare in advance for a full week of isolation; (ii) establishing a logistic and psychological support network to aid citizens who encounter unexpected needs; (iii) creating a dedicated app to issue exit permits only to members of the active cohort. The app in (iii) will not violate citizen privacy in any way, but only indicate if the device holder is in Cohort 1 or 2. Residents will be asked to present their app to enter shopping centers or public institutions. This can be done in addition to testing for symptoms, as already practiced in many countries.

Together, the proposed moderators create a framework that not only diminishes incentives for defection, e.g., by logistically supporting the isolated cohort but also eliminates the means, as, indeed, aside from daily outdoor strolling, practically all other out of home activities are automatically barred by the AQ framework itself. The strength of this implementation plan is that it achieves this without coercion, namely that almost no enforcement via

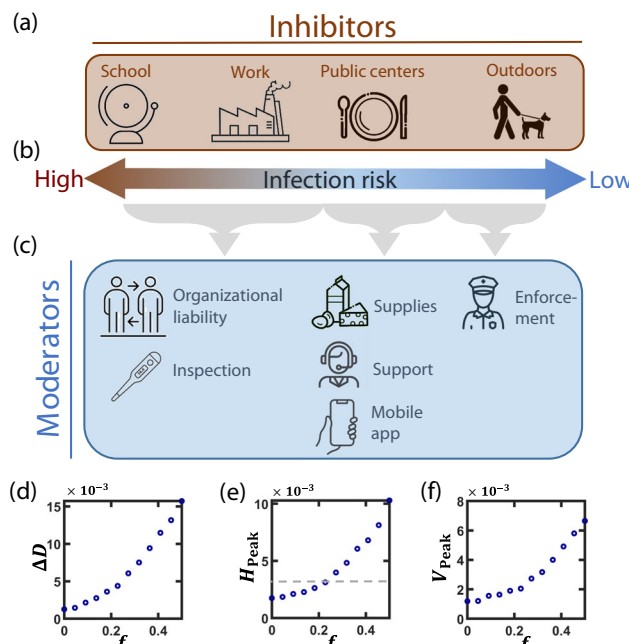

**Fig. 7 Driving social conformity for alternating quarantine. a** We identify four needs that inhibit potential cooperation: child care arrangements, work, purchasing supplies or services, and outdoor activities. **b** Infection risk is highest under extensive and continuous interactions, such as in school or at work, and least significant during open-air activities, such as strolling or exercising. We, therefore, focus on moderators mainly for the first three inhibitors. **c** To enhance social compliance we seek moderators that encourage conformity in lieu of coercive enforcement: school and work— due to their liability, schools, and workplaces will be naturally prohibited for the quarantined cohort, as both will be required to abide by the AQ routine, and therefore will not admit workers or students of the inactive cohort. In addition, routine inspections for symptoms will expose potential defectors who wish to conceal their infection. Public centers—we consider three moderators to deter individuals from seeking services or supplies: (i) instruct the population to obtain sufficient supplies in advance for a single week; (ii) establish a support network in case of unexpected needs; (iii) create a mobile app confirming an individual's cohort (1 or 2), that must be displayed upon entry to public centers. The outdoor activity could be moderated by enforcement, however, since it poses little infection risk, we believe such activity should, in practice, be ignored. **d** The residual mortality $\Delta D$ vs. the fraction of defectors/essential workers $f$, as obtained for the intermediate scenario $\beta = 0.20$, $\alpha = 0.15$. We find that AQ is robust under imperfect implementation, allowing to sustain a ~20% violation, either via formal exemption or by defection. **e, f** Similar results are obtained for $H_{Peak}$ and $V_{Peak}$; the average estimated national hospitalization capacity is indicated by the gray dashed line. Results represent an average of over 20 stochastic realizations on a network comprising 4000 households (~$10^4$ individuals).

authorized forces against individuals is required, maintaining a level of trust between citizens and government and securing personal freedoms. To complete the plan, at the end of the isolation week, all isolated residents will be required to report their health status via the app. Those who report symptoms, as well as their cohabitants, will remain at their stay-home status, going into isolation until their verified recovery.

Despite this detailed implementation plan, some level of violation of the AQ regime is unavoidable. This is either due to partial compliance, i.e. defectors, or because certain individuals hold essential positions and cannot leave their post for an entire week. Therefore, we now introduce a fraction $f$ of continuously

active individuals, defectors, or essential workers, who remain active at all times, both during their open shift as well as when their cohort is under quarantine. This $f$-fraction is extracted from the non-symptomatic ($S$, $E$, $I_{AS}$) or mild symptomatic ($I_M$) population, who may conceal their state. Excluded, however, are individuals experiencing severe symptoms ($I_S$, $I_C$, $H$, $V$) who, of course, remain isolated. Measuring our performance indicators, $\Delta D$ (3), $H_{Peak}$ (4), and $V_{Peak}$, we find that AQ can sustain defection/exemption up to $f \sim 0.2$, a 20% nonconformity level (Fig. 7d–f). Beyond that, we observe a significant decline in the strategy's performance.

## Discussion

The efficiency of the AQ strategy is rooted in three principles:

(i)   Partitioning the population into two cohorts reduces the volume of infectious interactions, comparable to a 50% quarantine (HQ).
(ii)  Working in weekly succession reduces the total duration of interaction within each cohort, similar to intermittent quarantines (IQ).

Combining these two factors together allows a similar net volume of socioeconomic activity as in any of these strategies, HQ or IQ, but with a multiplied mitigation effect. While (i) and (ii) are independent of the succession period, e.g., daily or weekly, our design of AQ around *weekly* alternations provides a third advantage:

(iii) It synchronizes the quarantine phase with the suspected incubation period of each cohort, hence systematically pruning out the invisible SARS-CoV-2 spreaders. Such synchronization can readily generalize to other infections, by accordingly tuning the AQ periodicity.

AQ can be implemented as an exit strategy, following a period of suppression via PWQ. As such, it allows a gradual reigniting of a dormant economy, while minimizing the risk of a recurring outbreak. However, our results indicate that it can also serve as a primary mitigation strategy, with comparable impact to that of a strict PWQ (Fig. 6). AQ should be further enforced with complementary measures, such as testing and selective protection of vulnerable populations (Supplementary Note 3).

A crucial strength of AQ is its robustness against defection, under some conditions withstanding as much as 20% violators. Nevertheless, we believe that the weekly relief, allowing people an outlet to continue their activity for half of the time, may, itself, increase cooperation levels. Indeed, while a complete lockdown is extremely stressful for the individual, the AQ bi-weekly routine relaxes the burden and may encourage compliance. Moreover, with workplaces and schools forced to operate in fully partitioned shifts, and with our suggested mobile app and logistic support network, the implementation of AQ has little dependence neither on self-motivation nor on externally enforced cooperation (Fig. 7). Indeed, schools and employment will naturally drive the population between activity and inactivity, with enforcement only required to treat outdoor recreation - which, in any case, has little contribution to the infection.

More broadly, we consider the fact that there is, inherently, some level of uncertainty regarding the disease parameters. We, therefore, examined the worst-case scenario, in which the infection rate during the active weeks is the same as that of the pre-mitigation spread. In practice, however, we expect many additional measures to be implemented in parallel to the quarantines, such as extended testing for infections, face-masks, and strict hygienic regulations at the workplace. At the least, we expect standard prophylactic behavior, such as avoiding contact or banning social gatherings, to be observed also during each cohort's active week. Such norms, which will continue until COVID-19 is fully eradicated, will further push down $\beta$, enhancing the effectiveness of our strategy even beyond the reported results.

Here, we have mainly discussed the epidemiological merits of AQ, and its implementation, in broad strokes, as a national strategy. In practice, different societies, as well as different economic sectors, will require specific adaptations. For example, while AQ is naturally compatible with non-professional industries, in which workers can be arbitrarily partitioned into shifts, it becomes more challenging in professional workplaces, where key personnel may be irreplaceable. Specific solutions, therefore, must be tailored to accommodate different economies and sectors. In light of AQ's unambiguous mitigating advantage, we believe such adaptations are, by far, worth the effort.

## Methods

**Smooth partitioning**. Assigning all individuals into cohorts may seem to require coordination that is difficult to scale at a national or regional level. Here, we offer a scheme to naturally achieve a smooth partition, minimizing both economic and individual stress • Employers. All employers will be allowed to resume activity, conditional on working in fully separated shifts. They will be given time to partition their workforce into two cohorts, $E_1$ and $E_2$, optimal for their business considerations. During this time, employers can also make other arrangements, such as training workers from $E_1$ to substitute for those in $E_2$, etc. • Local authority. Will inform all citizens of their cohort assignment, $R_1$ or $R_2$, based on, e.g., living address. Employers will update their lists $E_1$, $E_2$ accordingly • Conflict resolution. Conflicts can arise either due to employer needs or to individual preferences. For example, if an employer detects an unbridgeable discrepancy between the official assignment of a worker ($R_1$, $R_2$) and their professional needs ($E_1$, $E_2$). Similarly, an individual may wish to switch the cohort for personal reasons, e.g., to tend for a family member in the opposite cohort. To resolve such conflicts, citizens will be given the opportunity, until a preset date $T_f$ to file for transition, of their entire household, between $R_1$ and $R_2$. The local authority will update their lists accordingly, informing schools and other relevant institutions of the transition • Flexibility. The result is a friction-less scheme, essentially accommodating all transition requests. This is enabled thanks to the fact that AQ does not require a precise 50:50 partitioning. Hence, to allow a smooth and efficient split, both for the individual and for employers, the scheme is designed to be as flexible as possible • No micromanagement. By the deadline $T_f$ society will naturally be divided into two cohorts, in which all employees/employers are granted their ideal work schedule. The local authorities need not micro-manage this partitioning, just track it. Once the partition is set at $T_f$, no further transitions are enabled.

## Data availability

Empirical data on the number of infections in each country (Fig. 3) is available at https://www.arcgis.com/apps/opsdashboard/index.html. Data on hospital capacity can be found here: https://data.worldbank.org/indicator/sh.med.beds.zs Data on household size distribution was collected from https://population.un.org/Household/index.html/countries/840.

## Code availability

All codes to reproduce, examine, and improve our proposed analysis are available at https://github.com/drormeidan/ALDCOVID19.

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

## Acknowledgements

D.M. wishes to thank the support of the Bareket program in Bar-Ilan University, Israel. This research was supported by the Israel Science Foundation (grant No. 499/19), the Bar-Ilan University Data Science Institute grant for COVID-19 related research, and the Dangoor Center for Personalized Medicine at Bar-Ilan University.

## Author contributions

All authors designed the research. D.M. and N.S. conducted the data analysis and numerical simulations. D.M., R.C., S.H., N.S., and B.B. carried out the analytical derivations. R.S. advised on the virological aspects and E.Y. on the implementation. B.B. was the lead writer.

## Competing interests

The authors declare no competing interests.
