## [Peer Review File · Nature Communications]

Reviewers' Comments:

Reviewer #1:

Remarks to the Author:

This manuscript provides a model-based evaluation of a quarantine strategy for the control of COVID-19 epidemics.

The idea of an alternating quarantine is based on limiting as much as possible the transmission outside the household by optimizing the time lag between the active and quarantine phases. However, the model developed by the authors ignores the transmission between quarantined household members. Indeed, the model used in this manuscript is not suitable to test the effectiveness of the designed strategy; a more sophisticated model (e.g., agent-based simulations, network models) where transmission is explicitly simulated between household members and outside the household is needed (see detailed comments below). As such, my confidence in the results obtained by the authors is rather low. Moreover, the manuscript appears to be quite naïve (see for instance comments 2, 3, and 6).

1) The model ignores that asymptomatic individuals can transmit the infection to their household members during the quarantined phase (Equation 27 of the Supplementary Material). There are several studies showing relatively high (>20%) household secondary attack rates during the lockdown phase. If these individuals are then let free to infect in the community, they contribution to the overall transmission may be very relevant.

The model also ignores that detected symptomatic individuals who are isolated at home can transmit the infection to their (quarantined) household members. This may not be too relevant for the overall transmission of the virus as one can assume a strict testing policy of household members of detected symptomatic cases, but the authors are thus ignoring that the infections possibly generated by asymptomatic individuals add up to the toll for the healthcare system and, on the other hand, contribute towards the increase of the immunity in the population.

Overall, how all of these balances out it is hard to say and, indeed, requires to be tested by a proper model accounting for these factors.

This problem is evident also when the authors test the full quarantine strategy and they set $\beta=0$, thus neglecting the unavoidable household transmission.

2) The authors fit the reported number of deaths in the different countries, stating that "We use $D(t)$ since it represents an objective measure.". Actually, the number of deaths is the worst possible indicator exactly because of the lack of a common definition. COVID-19 is hardly the cause of death; deaths are associated with COVID-19 infection. Therefore, the definition of COVID-19-related death is extremely variable by country and, very often, by time period as well, due to the changing definitions used by the local authorities. Moreover, the delay between hospitalization and death is extremely variable over time, depending on the availability of ICU beds, medial personnel, improved knowledge on how to treat the patients, and so on.

3) The definition of the functions "P" (page 7 of the Supplementary Material) is crucial for the effectiveness of the proposed alternating quarantine strategy. There is an incredible amount of technicalities to keep into account when estimating key time-to-event distributions. First of all, how these data were obtained (I am not referring to what is the reference from which the data were retrieved, but to the protocol used in the epidemiological investigation), what was the sample size, whether the dataset was harmonized between locations, how censored data are accounted for, how missing data are considered, how multiple infectors are considered, and so on. These are all key details. However, the authors did not provide any detail. Overall, the analysis appears rather naïve. (Also the fact that no technical definitions are provided to each distribution and they are simply mentioned as "P" does not help to the overall feeling of the conducted analysis).

4) I understand the difficulty of defining R_0 for such a model, but it is hard to compare results to other studies and even between the baseline scenario and the mitigated ones without knowing the

R0 of the system.

5) Table II of the Supplementary Material. Either there is an issue with the lockdown dates or the table is not properly explained. For instance, Italy declared the lockdown before the UK (March 9 vs. March 23). However, in the table, it appears to be the contrary. In particular, in the UK (including England), the lockdown started on March 23, i.e., 61 days after Jan. 22 (not 46); in Italy, it started on March 9, 38 days after Jan. 22 (not 48).

6) Page 2, "Fortunately, lacking symptoms, such as coughing, which promotes virus shedding and dissemination, these asymptomatic individuals are likely less infectious than their symptomatic peers.". This is highly debated (see the debate between droplet vs. aerosol transmission). "Furthermore, asymptomatic individuals could very well have lower viral load in their respiratory tract and saliva [28–31]." This is highly debated as well, as there are several references showing that the difference is not statistically significant.

Reviewer #2:

Remarks to the Author:

Overall, the paper is timely and policy-relevant, the authors highlight an important issue that many countries are grappling with: What is the long-term strategy to deal with this to mitigate further transmission of SARS-CoV-2? Many countries across the world have implemented some form of lock-down where "normal" life grinds to a halt. The authors propose an alternating quarantine (AQ) strategy and employ mathematical projections that could help guide decisions to ease the lock-downs while still limiting social interactions. Even though eventually, pharmaceutical therapy or vaccine would be available, these non-pharmaceutical interventions will continue to have a role in interrupting the spread of SARS-CoV-2, locally, nationally and internationally.

Figure 1 provides a good overview of the strategy proposed by the authors. I also appreciate the authors' efforts to include "social defectors" or individuals who do not adhere to the recommendations.

The authors write: "Spreading the virus continues until the onset of symptoms, at which point the infected individuals enter isolation and cease to contribute to the spread." In the model, are symptomatic individuals also allowed to be "social defectors"? We have observed that while it is not possible to ensure that the individuals do not show any symptoms, and a considerable fraction would not disclose their actual symptoms (and may even attempt to conceal it) or adhere to the recommendations because their livelihoods are dependent on them working/resuming their activities.

The severity (and mortality) of COVID are heavily dependent on age and co-morbidity. How are the authors adjusting for these?

Many governments would probably be cautious as they slowly allow their residents to resume activities. Some may consider testing before allowing the virus-free or seroconverted and non-infectious individuals to continue activities. It would be interesting if the authors could also incorporate periodic or random screening of individuals using either serology and molecular tests.

It seems that in this model, the authors considered how the economy within a country could resume locally; however, have they factored in international travel?

As we observe the COVID-19 pandemic unfolding in many places, we also see vulnerable populations (such as those living in crowded accommodations, or nursing homes) are often overlooked when designing public health measures. I think this could be another limitation of the model proposed by the authors.

Reviewer #3:

Remarks to the Author:

I have reviewed the manuscript entitled Alternating quarantine for sustainable mitigation of COVID-19 by Barzel et al. In their study, the authors provide alternative strategies of containing COVID-19 outbreaks instead of a full population wide lockdown. In particular, they suggest a strategy in which the population is split into two cohorts, with each cohort alternating between quarantine for a specified amount of time. Their methodology consisted of a differential equation model taking into account several key disease characteristics including variable times for different disease stages and inclusion of presymptomatic phase and asymptomatic individuals. Overall the paper is well written, the model is well formed, and the key messages are clear.

I have a few comments:

1) It seems to me that the AQ strategy is a theoretical exercise and not practically feasible in a large population, with a high degree of spatial heterogeneity. To split and track two cohorts, it would require substantial resources from government, public health and private companies and seems to be a logistical nightmare. In large hubs like London and New York, there are thousands of companies that would have to comply and create scheduling that does not mix the two cohorts together. I am interested in hearing the authors comments on what their target population is? To me the AQ strategy seems entirely infeasible for large cities, but may be applicable for small, isolated "populations" such as those working in mines, oil rigs, and cruise ships.

2) The authors assume that asymptomatic individuals are not very infectious; however, studies have shown that asymptomatic individuals can be major drivers of disease spread and transmission. I am interested in seeing how the results would change if the relative infectiousness of asymptomatic individuals was higher than assumed.

3) Model parameters should be provided in a Table in the Supplementary including relevant citations and brief descriptions. I noticed that the authors provide the parameters in their online code files (on GitHub), but without a description, one gets lost in how the parameters are used. Providing a table will help with reproducibility.

I recommend this article for acceptance, provided the comments here are addressed.

To all Referees

We wish to thank you very much for your positive assessment of our contribution and for your thoughtful and constructive comments. We were especially excited by Referee 2 stating that our *paper is timely and policy-relevant... highlighting an important issue*, and by Referee 3 *recommending this article for acceptance to Nature Communications*, once comments are addressed. We are also grateful for Referee 1's through critique of our modeling framework, which prompted us to fundamentally redo our analysis, leading to, what we feel, is now a much stronger contribution.

Before we provide a detailed *point-by-point* response to *all* Comments, let us briefly overview the main changes introduced into the current submission:

Framing. While our application is tailored towards the current threat, the *idea* of alternating quarantine will remain relevant also beyond COVID-19. Therefore, focusing on COVID-19 as our prime, and indeed – most urgent – test case, we also discuss our strategy in the broader view of pandemic response.

Modeling. Our modeling is now based on a *stochastic temporal network framework*. This provides a dramatically more detailed account of the spread, going beyond *mean-field*, and therefore helping us capture the role of the network's degree distribution, and its temporal activity patterns. Most crucially, this framework allows us to account for the impact of in-house transmission, which – as noted by Referee 1 – is a highly relevant component in the context of any quarantine-based strategy.

Implementation. Approached by many governments, policy makers and industry leaders since our original submission, we have now gained much experience and a deeper understanding of the potential implementation challenges. Indeed, several industries and government branches have already implemented alternating quarantine, partially or fully, providing us with valuable experience upon which we can now build (Some examples were featured on media, *e.g.*, Leuze, Austria, Israel). Hence, in our current submission we include a more detailed Implementation section, covering practical aspects related to our strategy's application. Specifically, we explain how to most effectively and smoothly partition the population, garner cooperation and treat potential gaps in social compliance.

Synergistic measures. We discuss and test several *force-multiplying* policies that can (and should) be applied alongside our alternating quarantine strategy, from random testing to selective protection of vulnerable demographics.

These additional tests and our current more detailed modeling have truly strengthened our confidence in the suitability of alternating quarantine. Together with the additional, more minor changes, detailed in the report below, we hope the Referees will agree that our present submission should be communicated to the community as soon as possible.

Thankfully yours,
The authors

Reviewer #1

Comment 1

This manuscript provides a model-based evaluation of a quarantine strategy for the control of COVID-19 epidemics. The idea of an alternating quarantine is based on limiting as much as possible the transmission outside the household by optimizing the time lag between the active and quarantine phases.

However, the model developed by the authors ignores the transmission between quarantined household members. Indeed, the model used in this manuscript is not suitable to test the effectiveness of the designed strategy; a more sophisticated model (e.g., agent-based simulations, network models) where transmission is explicitly simulated between household members and outside the household is needed (see detailed comments below). As such, my confidence in the results obtained by the authors is rather low.

Response

We find this comment, together with its follow-ups in **Comments 2** and **3** to touch on an important issue that, we agree, was not properly treated in our original submission. We therefore wish to thank the Referee for prompting us to adjust our modeling framework, and improve our analysis. Indeed, the modeling that we employed in round one, while detailed in its account of the disease cycle (see **Fig. 1**), was perhaps highly aggregated and coarse-grained in its description of the social network structure - essentially assuming a well-mixed society (*i.e.* mean-field). This is, of course, simplistic and overlooks the social fine-structure, most crucially, as the Referee notes – the continuous interactions within household units. We have now, following this advice, constructed our model around a much more empirically relevant social network structure, as we detail below.

Temporal network model. Our current modeling preserves the detailed account of the disease cycle and its multiple compartments ($S, E, I_M, I_S, I_C, I_{AS}, H, V, R, M$), but now, instead of a mean-field implementation, we incorporate it on a stochastic temporal network environment:

- **Network structure (Fig. R1a).** To construct the network, we superimpose two separate graphs. First, the *external social network* A_{ij} , which captures the interaction patterns of individuals during their daily activity. This network includes all out of home interactions, *e.g.*, at work, public places or other social gatherings. The social network is characterized by degree distribution $P(k)$, which is often fat-tailed, capturing the level of heterogeneity in real-world social interactions. To err on the side of safety, in our analysis we examined two opposing scenarios – one in which $P(k)$ is bounded (Erdős-Rényi, Poisson, main text), and the other, where $P(k)$ is scale-free, accounting for the presence of hubs (**Figs. 2,3 in Supplementary Section 2**). For the purpose of AQ, we find no significant difference between these two extremes.

The second network, B_{ij} , represents *in-house* interactions, namely each individual is linked solely to his/her cohabitants. This network comprises a set of isolated cliques, ranging in size from $m = 1$ to 6, depending on the size of each household. Household size distribution $P(m)$ is extracted from empirical data pertaining to European societies, in which the average household (clique) is of size ~ 2.5 (**Supplementary Section 4.3**).

Superimposing these two network we obtain our complete network

$$G_{ij} = A_{ij} + B_{ij},$$

in which all out of home interactions are governed by A_{ij} and the in-house transmission occurs via B_{ij} .

- **Temporal interaction patterns (Fig R1c-d).** On top of this underlying G_{ij} nodes connect to each other stochastically, with periods of active interaction, *e.g.*, when i and j are collocated, and hence potentially infect each other, vs. intermittent periods, in which the i, j link remains idle. Therefore, for each link i, j we draw a random sequence of active and inactive time intervals, modeling the stochastic patterns of social contagion. For example, i interacts with j at an office meeting between 8:00 and 9:00 AM, then with k and m at their mutual work space from 9:00 to 11:00 AM, etc., hence first the G_{ij} link is activated for one hour, then G_{ik}, G_{im} and G_{km} are all simultaneously activated immediately after that for 2 hours. Such modeling accounts for the fact that the *sequence* of interactions, not just their duration, has a crucial effect on the pathways of the potential spread. It also captures the role of correlated interactions, for instance the fact that if i, k and i, m interact concurrently, then it is inevitable that also k, m interacted at the same time instance.

In a typical daily cycle, A_{ij} links are activated primarily during the day-time, while B_{ij} are reserved for the after-hours. Of course, if a household enters quarantine, either through one of the examined quarantine strategies (full, half, intermittent or alternating), or due to a household member showing symptoms, then B_{ij} becomes active also throughout the day. Therefore, the increase in secondary in-house transmission following quarantine is now properly accounted for.

- **Temporal parameters (Fig. R1b).** The intensity and the typical duration of all interactions depend on the nature of the links. Social links A_{ij} are characterized by an average daily activity time of T_1 , as explained above – predominantly during the day. Hence their stochastic activity/inactivity periods are drawn from a distribution, whose mean cumulative daily activity is T_1 . Cohabitant links B_{ij} , on the other hand, are characterized by average daily activity T_2 , concentrated mainly in the night hours. Typically we expect that $T_2 > T_1$, capturing the fact that cohabitants spend more time together in potentially infectious interaction than, *e.g.*, office mates.

Together, our model simulates a 150 day social scenario at a 15 minute resolution, in which individuals interact temporally both out of home (A_{ij}, T_1) and in house (B_{ij}, T_2), in a sequence of stochastically generated daily cycles.

Figure R1. Stochastic temporal network framework. (a) The social network, including external links A_{ij} and in-house links B_{ij} . (b) Each of these networks, A_{ij} and B_{ij} is characterized by its structural and temporal parameters. From these we extract the in-house transmission rate α and the growth rate β . (c) – (d) A typical daily cycle. In each 15 minute instance a link can be active (dark) or inactive (light). Infections can occur along active links. A_{ij} is activated mainly during work-hours, B_{ij} in the after-hours. When under quarantine only B_{ij} is active (unless i or j are violators).

- **Infection parameters (Fig. R1b).** Infection across the i, j link may take place whenever the link is active. We denote the average infection probability per unit time as p_1 for social links (A_{ij}) and p_2 for household links (B_{ij}). Also here, under typical conditions, $p_2 > p_1$, as, indeed, cohabitants, often family members, interact more extensively and physically than social peers. Consequently, the average out-of-home infection rate is $\rho_1 = p_1 T_1$, and the average in-house infection rate is $\rho_2 = p_2 T_2$, providing two independent model parameters by which to control the evolution of the model, and the relative contribution of in-house vs. external viral transmission.
- **Infection growth rate and in-house infection rate.** While many of the relevant parameters are empirically accessible, for instance the disease rate-constants (Fig. 1) or the household size distribution $P(m)$, others, such as ρ_1, ρ_2 are unknown. We, therefore, examine different spreading and mitigation scenarios by scanning different values of the parameters ρ_1 and ρ_2 , in each case - averaging over a set of independent realizations. These *natural* model parameters determine two *observable* quantities that provide direct insight on the patterns of spread:
 - (i) **Infection growth rate β .** The exponential proliferation of infections $I(t) \sim e^{\beta t}$, captures the empirically observed growth in infections at the early stages of the pandemic. This parameter can be set by tuning the combined effect of the in-house (ρ_2) and out-of-home (ρ_1) transmission. Similar to our original manuscript, also in our present submission, we cover a range of potential rates β - capturing worst, intermediate and best case scenarios - thus accounting for the inherent uncertainty in the actual value of this parameter.
 - (ii) **In-house transmission rate α .** This parameter captures the fraction of infections that occurred via household links. For each realization we extract α directly from the simulated spread and use it to quantify the relative contribution of in-house vs. out-of-home transmission to the spread. As α approaches unity ($\rho_2 \gg \rho_1$), we enter a regime in which the majority of infections occur within households. Conversely, in the limit $\alpha \rightarrow 0$ ($\rho_1 \gg \rho_2$) we have a negligible contribution of intra-family transmission to the spread, as almost all infections occur via the external social network A_{ij} .

With this significantly more detailed modeling framework, we can examine the performance of all our examined mitigation strategies, AQ, IQ, HQ etc., under varying levels of household secondary transmission, capturing both the merits, and the potential adverse effects ingrained in all the quarantine-based strategies.

- **Uncertainties.** This detailed modeling framework requires, as input, several parameters and assumptions, some attainable from data, others - unknown. For example, the disease time-scales and transition probabilities were all tuned from available biological measurement, hence they are *known*. Similarly, the daily routine (time spent in and out of home) and the household sizes were also attuned to real data. To treat the unknown parameters, we systematically examined a spectrum of possible scenarios. For example:
 - (i) **Infection growth rate β .** While we can evaluate β from data collected on the early stages of the spread (Fig. 3), it is difficult to assess how β changes in response to people's adaptive behavior. We, therefore, include best, intermediate and worst case scenarios to capture the unknown effect of these adaptive prophylactic norms.
 - (ii) **In-house transmission rate α .** Estimates for the fraction of transmissions occurring in-house vary, and hence, also here, we examine different scenarios.

Together, we test the different mitigation strategies under low, medium and high rates of in-house transmission.

- (iii) **Social network A_{ij}** . Using both an Erdős-Rényi (**Fig. 4**) and a scale-free (**Supplementary Section 2**) network we examine two highly distinct patterns of social contagion.
- (iv) **Cooperation f** . Similar to our original submission we vary the fraction of defectors/formally exempt individuals to observe the strategy's robustness under a spectrum of social conformity levels (**Fig. 8**).

Taken together, our analysis is now significantly more realistic and comprehensive, accounting for a range of parameter settings and model assumptions. Most crucially, it allows us to thoroughly examine the effect of in-house secondary transmission, as suggested by the Referee.

Comment 2

The model ignores that asymptomatic individuals can transmit the infection to their household members during the quarantined phase (Equation 27 of the Supplementary Material). There are several studies showing relatively high (>20%) household secondary attack rates during the lockdown phase. If these individuals are then let free to infect in the community, they contribution to the overall transmission may be very relevant. The model also ignores that detected symptomatic individuals who are isolated at home can transmit the infection to their (quarantined) household members. This may not be too relevant for the overall transmission of the virus as one can assume a strict testing policy of household members of detected symptomatic cases, but the authors are thus ignoring that the infections possibly generated by asymptomatic individuals add up to the toll for the healthcare system and, on the other hand, contribute towards the increase of the immunity in the population. Overall, how all of these balances out it is hard to say and, indeed, requires to be tested by a proper model accounting for these factors.

Response

As detailed above, such secondary transmission between household members is now incorporated into our modeling framework. The effect suggested by the Referee, of asymptomatic individuals infecting their household members, while certainly relevant, seems, in our simulations to be of minor impact on AQ's performance. Interestingly, we find that to some extent, this effect can actually *mitigate* the harmful consequences of asymptomatic transmission.

To understand this, consider an asymptomatic individual i infected towards the end of Week 1, entering quarantine in Week 2, and then – having no symptoms, goes on to infect others in Week 3. Such sequence of events is enabled according to the AQ routine, and therefore may contribute at some level to the spread. However, if i lives with other family members, it is likely that he/she will infect them, as they spend a full week at home together during Week 2. This is, of course, a negative outcome of the quarantine, however it also has an upside: indeed, most chances are, that at least one of the infected cohabitants *will* develop symptoms. And once that occurs, the entire household enters quarantine until all members are cleared. Hence, surprisingly, in this scenario the in-house transmission, has a mitigating effect, rather than an exacerbating one.

To evaluate the impact of this effect we measured the fraction of asymptomatic individuals that, despite having no symptoms, were still in quarantine during their active week. According to the AQ routine, they are, in principle, allowed to remain active. Hence, the only reason for them to be isolated is due to having a symptomatic member in their household. Indeed, we find that, on average, 30% of asymptomatic individuals within the active cohort were, in fact, under quarantine. These 30%, if it were not for the in-house transmission, would have been socially active, spreading the virus.

This is an interesting side-effect of the quarantine that we would have overlooked, had we not adopted our revised modeling framework. We now added a discussion along the lines of this response in the Discussion section of the paper.

Comment 3

This problem is evident also when the authors test the full quarantine strategy and they set $\beta = 0$, thus neglecting the unavoidable household transmission.

Response

Of course, the Referee is correct. Even under Full quarantine (FQ) infections between cohabitants continue for some time. Fortunately, prompted by the Referee's other comments (1 and 2) we have now fundamentally changed our modeling framework, to include also in-house transmission. Therefore, when we now employ Full quarantine, the effect of secondary infections within households is no longer neglected – providing a fair baseline for comparison.

Comment 4

The authors fit the reported number of deaths in the different countries, stating that “We use $D(t)$ since it represents an objective measure.” Actually, the number of deaths is the worst possible indicator exactly because of the lack of a common definition. COVID-19 is hardly the cause of death; deaths are associated with COVID-19 infection. Therefore, the definition of COVID-19-related death is extremely variable by country and, very often, by time period as well, due to the changing definitions used by the local authorities. Moreover, the delay between hospitalization and death is extremely variable over time, depending on the availability of ICU beds, medical personnel, improved knowledge on how to treat the patients, and so on.

Response

We wish to thank the Referee for this comment. We now use the confirmed infections $I(t)$ as our source for parameter estimation. Quite encouragingly, we find, despite the unavoidable differences between countries, that the extracted parameter β , characterizing $I(t)$'s exponential growth, is, generally, consistent, across our 12 independent destinations.

We also wish to emphasize that, given the inherent uncertainty on the value of β , we specifically test all our competing strategies, HQ, IQ, FQ etc. in **Figs. 4** and **7**, under variable scenarios, assuming large, intermediate or small infection rates β (this is in addition to a range of household transmission rates α , see our response to **Comment 1**). Indeed, β is highly affected by people's behavior, with simple practices, such as personal hygiene, physical distancing and application of protective masks, potentially changing its value dramatically. Therefore, even if β is extracted from data, be it mortality or infection,

it is practically impossible to truly know its value, now that society has become more cautious and adapted its social behavior. Examining a range of β values, best-case, intermediate and worst-case scenarios, we ensure that our findings are insensitive to discrepancies in this elusive parameter.

Comment 5

The definition of the functions “P” (page 7 of the Supplementary Material) is crucial for the effectiveness of the proposed alternating quarantine strategy. There is an incredible amount of technicalities to keep into account when estimating key time-to-event distributions. First of all, how these data were obtained (I am not referring to what is the reference from which the data were retrieved, but to the protocol used in the epidemiological investigation), what was the sample size, whether the dataset was harmonized between locations, how censored data are accounted for, how missing data are considered, how multiple infectors are considered, and so on. These are all key details. However, the authors did not provide any detail. Overall, the analysis appears rather naïve. (Also the fact that no technical definitions are provided to each distribution and they are simply mentioned as “P” does not help to the overall feeling of the conducted analysis).

Response

Our construction of the COVID-19 disease cycle was carried out in two steps:

- I. Mean transition times.** First, we evaluated the average transition times. These have been tested, quite extensively, across hundreds of papers, and in different locations and settings. The picture that emerges condenses rather narrowly around the time-scales that we provide in our disease cycle, which are congruent with the major publications that have been circulated on SARS-CoV-2.
- II. Distribution of transition times.** To challenge our AQ strategy it was important that we also account for the variability in these transition times. Especially in a few specific transitions with direct impact on AQ’s performance, such as the duration of the pre-symptomatic phase, or the time for asymptomatic individuals to recover.

While Step I relies on extensive empirical data, in Step II, due to the novelty of SARS-CoV-2, we had to fill several gaps. First, decide on the *family* of relevant distributions, then choose the specific distribution parameters. For the family of the distribution, we selected a Weibull distribution, based on the following considerations:

- This distribution is characteristic of the incubation and recovery times of several respiratory viruses, such as influenza virus, adenovirus, and the human coronaviruses MERS-CoV and SARS-CoV. Together, this provides firm grounds for such distribution to be the *best guess* also for SARS-CoV-2. Indeed, all previous viruses mentioned above have been extensively studied over the course of several years, on large samples, representative of different populations and countries, and hence the Weibull family represents a natural candidate also for COVID-19. Other similar distribution families, such as the Gamma distribution, are equally relevant, but for all practical purposes, are indistinguishable in the present context from Weibull.
- Few preliminary studies have directly indicated that Weibull also plays a role in COVID-19. These studies are still of limited size, *e.g.*, 88¹, 158² and 181³ patients, and

¹ *Euro Surveill.* **25**(5), 2000062, 2020. DOI:10.2807/1560-7917

² *J Clin Med.* **9**(2), 538, 2020. DOI: 10.3390/JCM9020538

³ *Ann Intern Med.* **20**, 504, 2020. DOI: 10.7326/M20-0504

are also geographically centered around Wuhan. Therefore, on their own they are insufficient to confirm Weibull as the single relevant distribution family, but certainly reinforce it as a prime candidate.

- As a rule, similar to all uncertainties we faced, our approach was to *err on the side of safety*. Here, increasing the variability of, *e.g.*, the incubation period, introduces additional challenges to AQ. Indeed, AQ becomes perfect in the limit where everyone has an exact 5 day pre-symptomatic phase, and is compromised by the tail of individuals with a potentially extended pre-symptomatic phase. Therefore, even if inaccurate, selecting a long-tailed distribution, such as Weibull, places AQ at a strategic disadvantage, in a sense helping us examine its performance under adversarial conditions.

We wish to emphasize that the distribution *family* is often an intrinsic characteristic of the system, insensitive of specific details. For example, the Normal distribution being a fingerprint of randomness, or the Geometric distribution capturing memory-less processes. The consistent emergence of Weibull/Gamma distributions in the context of our immune response to viruses, indicates that it is a relevant family also here. Therefore, together with the preliminary indications obtained for SARS-CoV-2, and the fact that it is a worse-case assumption in the context of AQ, we believe it represents a safe choice for our disease model.

Once converging around the Weibull family, what is left is to evaluate its parameters. This can be done based on highly reliable estimates of the average and median transition times for the different disease pathways, as we explain in **Supplementary Section 4.1**.

Comment 6

I understand the difficulty of defining R_0 for such a model, but it is hard to compare results to other studies and even between the baseline scenario and the mitigated ones without knowing the R_0 of the system.

Response

Our compartmental model is significantly more elaborate than the standard SIS/SIR models, and it is therefore, as the Referee acknowledges, difficult to evaluate R_0 . Instead, we follow the common track and quantify the severity of the spread using the initial daily growth rate of the epidemic. For small t , we assume $I(t) \sim e^{\beta t}$, and use the observed exponential growth rate β to parameterize the proliferation of the virus. Using data from 12 countries, we find that, unmitigated, $\beta \approx 0.26$. This observed value is in agreement with many other publications on COVID-19, indicating that it is a relevant baseline scenario of the unmitigated spread. For example, Bar-on *et al.*⁴ estimate $\beta = 0.23$, and similar rates also appear in Wu *et al.*⁵ and Li *et al.*⁶.

Comment 7

Table II of the Supplementary Material. Either there is an issue with the lockdown dates or the table is not properly explained. For instance, Italy declared the lockdown before the UK (March 9 vs. March 23). However, in the table, it appears to be the contrary. In particular, in

⁴ eLife 2020;9:e57309 DOI: [10.7554/eLife.57309](https://doi.org/10.7554/eLife.57309)

⁵ Lancet 2020; 395(10225):689-697. DOI: [10.1016/S0140-6736\(20\)30260-9](https://doi.org/10.1016/S0140-6736(20)30260-9)

⁶ N Engl J Med 2020; 382:1199-1207 DOI: [10.1056/NEJMoa2001316](https://doi.org/10.1056/NEJMoa2001316)

the UK (including England), the lockdown started on March 23, i.e., 61 days after Jan. 22 (not 46); in Italy, it started on March 9, 38 days after Jan. 22 (not 48).

Response

Thanks for detecting this. The table has now been corrected. We have also added multiple additional Tables to the SI, summarizing all our parameters and estimates.

Comment 8

Page 2, “Fortunately, lacking symptoms, such as coughing, which promotes virus shedding and dissemination, these asymptomatic individuals are likely less infectious than their symptomatic peers.” This is highly debated (see the debate between droplet vs. aerosol transmission). “Furthermore, asymptomatic individuals could very well have lower viral load in their respiratory tract and saliva [28–31].” This is highly debated as well, as there are several references showing that the difference is not statistically significant.

Response

We agree. And we have now removed these statements in the revised text. However, we wish to clarify that *at no point did our analysis rely on these assumptions that asymptomatic individuals are less infectious*. As we stated in our original text, and now reiterate more clearly

... to err on the side of safety, in our modeling of the spread we use a uniform infection rate, for all individuals - symptomatic or asymptomatic...

Hence, in our modeling, both originally and at present, we assume, explicitly, that asymptomatic individuals are *just as infectious as their symptomatic peers*. Therefore, the doubts mentioned by the Referee, while indeed correct, have no bearing on the reported results. *We are aware that asymptomatic individuals are the potential Achilles heel of AQ, and therefore, take special care not to adopt any relaxing assumptions pertaining to their infectiousness.*

We once again wish to thank the Referee for his/her thorough, deep and thoughtful review of our contribution. The comments, indeed, prompted us to fundamentally revise our modeling, leading to, what we believe, is a significantly strengthened paper.

Reviewer #2

Comment 1

Overall, the paper is timely and policy-relevant, the authors highlight an important issue that many countries are grappling with: What is the long-term strategy to deal with this to mitigate further transmission of SARS-CoV-2? Many countries across the world have implemented some form of lock-down where “normal” life grinds to a halt. The authors propose an alternating quarantine (AQ) strategy and employ mathematical projections that could help guide decisions to ease the lock-downs while still limiting social interactions. Even though eventually, pharmaceutical therapy or vaccine would become available, these non-pharmaceutical interventions will continue to have a role in interrupting the spread of SARS-CoV-2, locally, nationally and internationally.

Response

We wish to thank the Referee for this concise summary of our contribution. Indeed, social distancing schemes will continue to impact our socioeconomic routine in the foreseeable future, until the development of a therapeutic/vaccine. Therefore, it is *now* the time to consider more resolved alternatives to the Pavlovian lock-down response. Alternating quarantine, it seems, currently offers the ideal balance between mitigation efficiency (reducing R_0 by ~ 4) and economic activity (50% active at all times).

We add, that even posterior to the development of a vaccine, AQ will remain relevant as a framework to treat potential future pandemics. There is, we believe, little reason to assume that COVID-19 is our last public health crisis.

Comment 2

Figure 1 provides a good overview of the strategy proposed by the authors. I also appreciate the authors’ efforts to include “social defectors” or individuals who do not adhere to the recommendations.

Response

Thanks.

Comment 3

The authors write: “Spreading the virus continues until the onset of symptoms, at which point the infected individuals enter isolation and cease to contribute to the spread.” In the model, are symptomatic individuals also allowed to be “social defectors”? We have observed that while it is not possible to ensure that the individuals do not show any symptoms, and a considerable fraction would not disclose their actual symptoms (and may even attempt to conceal it) or adhere to the recommendations because their livelihoods are dependent on them working/resuming their activities.

Response

That is a good point. In our original formulation we only allowed defection among the pre-symptomatic or asymptomatic. Our assumption was that most people will not have the audacity to violate the rules so bluntly as to remain active while knowingly contagious. This assumption, we emphasize, had, anyway, little impact on our results as our analysis

was *comparative* in nature. Hence, we assumed similar cooperation levels also in all other competing strategies that we considered: Unmitigated (UM), Intermittent quarantine (IQ) and Half quarantine (HQ). Indeed, as we only *compare* between different forms of quarantine, what matters is the *fairness of the comparison*, i.e. that it is carried out under similar assumptions.

With that said, following this comment we decided to allow also defection among the mild symptomatic (I_M). Clearly the severe and critical, who are hospitalized, cannot violate the isolation or conceal their symptoms. The outcome, presented in **Fig. 8** of the revised submission, continues to indicate AQ's robustness against partial compliance.

Our current inclusion of symptomatic defectors is a worst case assumption, as, indeed, many countries are now testing for symptoms, such as fever, upon entry to schools, workplaces or public institutions. It is, therefore not easy to conceal even mild symptoms. Still, we agree with the Referee, that testing our strategy, it is better to err on the side of safety, as we do, in our paper, with all instances of modeling uncertainty.

Comment 4

The severity (and mortality) of COVID are heavily dependent on age and co-morbidity. How are the authors adjusting for these?

Response

We wish to thank the Referee for this suggestion, which prompted us to add this factor into our modeling framework. Let us first begin, though, with our doubts regarding the implementation of such a differential disease cycle, and the course we have decided to take following their consideration.

The disease cycle shown in **Fig. 1** captures the *average* evolution of COVID-19, as extracted from multiple studies. In reality, as the Referee correctly notes, across the population, there are differences in the cycle, related to risk factors, such as age and co-morbidity. The challenge is that these discrepancies are not fully mapped at present, and we fear that wrongly estimating the disease parameters for the different sub-populations, can potentially introduce uncertainties and errors into our analysis. Therefore, we now introduce two separate models:

- In the main text we use the aggregated disease cycle, which represents the most reliable description of the currently mapped COVID-19 transition rates and associated probabilities.
- In **Supplementary Section 3** we also examine a differential disease cycle, in which we divide the population into

Figure R2. Protecting the vulnerable population. (a) Hospitalization $H(t)$ vs. t under HQ (red), IQ (turquoise) and AQ (blue) without selective isolation. (b) Adding selective isolation of vulnerable population reduces hospitalization under all schemes. (c) Peak hospitalization H_{Peak} with (dark) and without (light) selective protection. (d) – (f) Similar analysis for ventilation $V(t)$, and (g) – (i) mortality $D(t)$. In all cases, selective isolation is beneficial. Under AQ, the effect is less pronounced as all the indicators are very low to begin with.

healthy (80%) and vulnerable (20%) individuals, capturing, *e.g.*, the higher morbidity among the elderly.

Such differentiation is favorable for the performance of AQ, since it allows us to selectively isolate the vulnerable population – as, indeed, practiced in most countries these days. Moreover, keeping the *average* disease cycle conserved, the introduction of a vulnerable sector, renders the cycle for the remaining population less severe. For example, in the average disease cycle we have 5% of the exposed (E) individuals becoming critically ill (I_C). If, however we introduce, say, a 10% vulnerable population who reach I_C with a 14% probability, then, in order to sustain the overall 5% average, we must set the transition probability of the remaining 90% healthy individuals at 4% ($0.1 \times 14\% + 0.9 \times 4\% = 5\%$). The net result will be, that if we isolate the vulnerable population, the remaining 90% will exhibit lower hospitalization, ventilation and mortality rates.

In that sense, our use of the single disease cycle in the main text represents a cautious modeling assumption, helping us examine AQ under more challenging conditions.

Our newly added analysis in **Supplementary Section 3** indicates, as expected, that adding special protection of the sensitive population, can greatly contribute to reducing severe and fatal case counts, as quantified by the peak hospitalization and ventilation H_{Peak} and V_{Peak} , and by the residual mortality ΔD (**Fig. R2**). This, however, is not unique to AQ, and is equally beneficial under the alternative contenders, IQ and HQ. Therefore, without doubt, such protection is highly recommended as a complement to any mitigation strategy, as we now explicitly discuss in the paper's **Discussion** section.

Comment 5

Many governments would probably be cautious as they slowly allow their residents to resume activities. Some may consider testing before allowing the virus-free or seroconverted and non-infectious individuals to continue activities. It would be interesting if the authors could also incorporate periodic or random screening of individuals using either serology and molecular tests.

Response

We agree – any successful mitigation strategy should be supported by testing, contact tracing as well as any other measures to detect and isolate infectors. We therefore, now added such component to our analysis in the newly added **Fig. 6**. Wishing to maintain our focus on the main theme of our present contribution – the AQ strategy – we followed the Referee's advice and incorporated the most basic strategy of periodic and random screening – thus avoiding, in the present context, more elaborate and optimized testing strategies, whose characterization manifests an independent research topic on its own right.

We emphasize that, ideally, AQ should be accompanied by a *smart* testing strategy, aiming, under limited testing capacity, to detect the maximal number of infected (or potentially infected, *a-lá* contact tracing or digital tracking) individuals. Developing such optimized testing protocols is discussed in many dedicated works, but, we feel, that here it may divert the focus of our current message.

Still, as we discuss below, we found an interesting aspect related to the testing efficiency, rooted in AQ's partitioning of the population. We therefore thank the Referee for prompting us to examine this complement to AQ.

Testing and AQ. Under the suggested framework of random testing, we quantify our testing capacity χ via the fraction of the population that can be screened within a single week. We find that there are two limiting cases. In case $\chi \ll 1$, such random testing provides negligible benefit. Indeed, under AQ, the number of infected individuals at any point in time is low, of the order of $\sim 10^{-3}$, hence, coupled with a small χ , very few tests will end up positive. Consequently, detection rate is too low to have a meaningful impact. In this limit, therefore, one must employ a *smart* testing strategy, whose design is unrelated to AQ.

As χ approaches 50%, however, it can truly enhance the efficiency of AQ, thanks to our strategy's natural partition of the society. The idea is to direct all testing in each week to the quarantined cohort. As this cohort is isolated at home, its state is *frozen*, and hence one can spread the tests across the week to cover the entire quarantined population. If, indeed, all the quarantined population is tested in, say, Week 1, this guarantees an almost infection-free workforce resuming activity in Week 2. Continuing this testing routine on the second cohort during Week2 ensures a clean active society by the end of the first AQ cycle.

Hence, within 1 to 2 weeks we fully annihilate all out-of-home infections – arriving at a state which can be practically equated with Full quarantine (FQ). This is despite the fact that 50% of the population continues to remain active at all times. Indeed, **Fig. 6c** shows that AQ with $\chi \approx 0.5$ exhibits a similar decay to FQ, with a delay of 1 – 2 weeks (10 days). Such efficiency is directly related to AQ's breakdown of society into two separate cohorts – allowing us to focus our testing on the quarantined population, a benefit lacking in, *e.g.*, IQ. Of course, as we consider lower χ , this effect is weakened. Still, the fact that at any point in time we can selectively direct our testing towards the relevant 50% of the population, adds to the mitigation gain obtained per each test.

The relevant limit of χ . At present, most countries are at the $\chi \ll 1$ limit, with a weekly testing capacity that is significantly smaller than the population. Under such conditions the suggested random cross-population testing becomes irrelevant. However, an inexpensive home operable testing kit is likely to be on the market within the foreseeable future. If/when such tests become available, they can become a force-multiplier to AQ – further pruning the infected population in each weekly cycle.

We wish to thank the Referee for prompting us to include this discussion in the present submission.

Comment 6

It seems that in this model, the authors considered how the economy within a country could resume locally; however, have they factored in international travel?

Response

International travel can be readily introduced into our modeling framework, in the form of a constant influx/outflow of individuals, of whom some are exposed (E) or infected (I_{AS} or I_M , we assume that severe or critically ill cannot travel). We find, however, that the effect of such mobility is negligible, to the extent that it becomes undetectable in our simulations. To understand this, we must consider the scale of international mobility. Using flight record data collected over the past decade, *i.e.* pre-COVID-19, we estimate that the relevant influx of individuals into a typical destination is of the order of $\sim 10^{-3}$. Of these, only a small margin are infected, and hence the endogenous infections, occurring

within each local society by far exceed the contribution of incoming travelers. To be clear, international travel plays a crucial role in the initial penetration of the disease into an infection-free country. But once the local spread is instigated, it seems to have a negligible effect. We, therefore, avoided this factor, so as to not further complicate our modeling.

Comment 7

As we observe the COVID-19 pandemic unfolding in many places, we also see vulnerable populations (such as those living in crowded accommodations, or nursing homes) are often overlooked when designing public health measures. I think this could be another limitation of the model proposed by the authors.

Response

We wish to thank the Referee for this constructive comment. Our AQ strategy, as we now emphasize, is not standalone. It can, and should, be combined with additional steps to maximize its efficiency. This includes, quite naturally, standard prophylactic measures, such as personal hygiene, physical distancing and protective masking. But, in reality, we certainly recommend to couple AQ also with selective protection of vulnerable populations, extended testing (as suggested in **Comment 5**), geographic partitioning and other policies to minimize economic impact and enhance mitigation efficiency. In that sense, AQ is not mutually exclusive with other policies, the contrary – it should be viewed as a *force-multiplier*, reinforcing and reinforced by all other relevant mitigation efforts.

Fortunately, our current modeling framework, based on complex temporal networks is highly flexible and allows us to introduce intricate social fine-structure. For example, distinguishing between cohabitant links, *e.g.*, family members, and social links, such as coworkers (as we now do in our new modeling). In this framework we can naturally introduce sub-populations, such as elderly people that are at high-risk, and examine the effect of their selective isolation. As explained in our Response to **Comment 4** above, we include such examination in the newly added **Supplementary Section 3**.

Indeed, as the Referee suggests, we observe that such selective isolation of the vulnerable population is crucial in reducing hospitalization, critical care and potential mortality. Hence, in addition to the analysis in the Supplementary material we also added a dedicated discussion in the main text on the importance of complementary measures to AQ, from treating the vulnerable population to implementing a coordinated testing strategy.

Reviewer #3

Comment 1

I have reviewed the manuscript entitled Alternating quarantine for sustainable mitigation of COVID-19 by Barzel et al. In their study, the authors provide alternative strategies of containing COVID-19 outbreaks instead of a full population wide lockdown. In particular, they suggest a strategy in which the population is split into two cohorts, with each cohort alternating between quarantine for a specified amount of time. Their methodology consisted of a differential equation model taking into account several key disease characteristics including variable times for different disease stages and inclusion of pre-symptomatic phase and asymptomatic individuals.

Overall the paper is well written, the model is well formed, and the key messages are clear.

Response

We wish to thank the referee for this concise and positive summary of our work. Our modeling, indeed, accounts for the detailed COVID-19 disease cycle, as noted by the Referee. In the revised submission offered here, it also includes a more accurate description of the social network, using a stochastic temporal network framework, which allows us to capture the impact of (i) in house vs. out-of-home transmission; (ii) the effects of the social network degree distribution; (iii) the impact of the temporal patterns of the infectious interactions.

I have a few comments:

Comment 2

It seems to me that the AQ strategy is a theoretical exercise and not practically feasible in a large population, with a high degree of spatial heterogeneity. To split and track two cohorts, it would require substantial resources from government, public health and private companies and seems to be a logistical nightmare. In large hubs like London and New York, there are thousands of companies that would have to comply and create scheduling that does not mix the two cohorts together. I am interested in hearing the authors' comments on what their target population is? To me the AQ strategy seems entirely infeasible for large cities, but may be applicable for small, isolated "populations" such as those working in mines, oil rigs, and cruise ships.

Response

We wish to thank the Referee for this comment, prompting us to further deepen our discussion on AQ's practical implementation. Before we answer in detail, let us first say that, since the AQ idea was made public, around the time of our original submission, we were invited to introduce it in front of numerous government committees, representatives of business sectors around the world, and policy makers. During these discussions we were exposed to many issues related to the practicalities of the strategy, and had the opportunity to rethink many aspects of its potential implementation. The lessons we have learned have now been incorporated into the **Implementation section** of the paper, which we believe is, now, much stronger.

We treat an array of potential challenges, including, but not limited to the ones mentioned by the Referee. *Specifically, we base our implementation on the notion that no coordination at a regional or national level is required. Indeed, as the Referee correctly states, such micro-*

level centralized planning of individual scheduling is completely impractical. Hence, the partitioning has to work in a decentralized fashion, which as we explain below, is significantly easier than it might seem at first glance.

Small scale application

While AQ is most effective as a national strategy, we agree with the Referee that it can also be considered at a smaller scale, *e.g.*, at a regional level, or in large industrial corporations or factories. Applying AQ at these levels is not perfect, as workers may continue to cross-infect through their off-work interactions, unless they are truly isolated, as in *mines, oil-rigs or cruise ships*. Still, such local application, which is certainly within practical reach, will continue to minimize infections within the workplace, be it a factory, a warehouse or a shared office space. In a scenario where social gatherings are banned or, at least, reduced, workplace infections play a significant role in the spread, and hence any company or corporation that opts into AQ can contribute to the mitigation effort.

Let us emphasize that such implementation is by no means a theoretical exercise. It has, in fact already been practiced by several corporate organizations, as well as government branches (in Israel) and schools (in Israel and Austria) - all following our original publication of the method on the *arXive*. Indeed, most of these early adopters have directly consulted us, and some have also been covered by media, for example • Leuze jumps on alternating-quarantine concept to keep business on track • Austria will reopen schools with split classes next month.

Such small scale application can become widespread, if following our paper, appropriate incentives are offered to businesses that wish to resume activity in this manner. Many countries have already detailed a standard for resuming business activity. This standard can (and should) include a component of AQ – as part of the conditions for opening the business. If not as a prerequisite, at least as a preferred option. If, indeed, many businesses opt in to this scheme, the overall effect can be a significant, non-coercive and decentralized, suppression of work-related infection.

However, the idea must be communicated for such incentives to be incentivized...

National level application

The above scenario illustrates that businesses are key to the successful application of AQ, and therefore we tailor our implementation around their needs and liabilities. Below we detail a scheme for a smooth assignment of individuals into cohorts under a *national level implementation of AQ*. The scheme is designed around two principles: (i) requiring no national level micro-management; (ii) aiming for the most convenient partition for the individuals/businesses, thus enhancing cooperation and socioeconomic prosperity.

Business liability. Employers will be instructed to resume their activity, conditional on working in shifts. Hence, each employer will consider the optimal partition for their business functionality, including, if needed, training employees to substitute for each other, or to work from home during their quarantined week. For some businesses, such arrangements may be sub-optimal, of course, but still, we assume, better than complete inactivity. Note, that no coordination between the businesses is required, as each employer designs their own lists, E_1 and E_2 , of employees they desire to be in Cohorts 1 and 2, respectively.

Households. Independently of the lists constructed by the businesses, the local authority informs its citizens of its partition R_1 and R_2 of residents in each cohort. This list is based on each individual's living address, ensuring all cohabitants are in the same cohort.

Citizens are instructed to follow their schedule according to R_1 and R_2 , and schools are instructed to admit students accordingly.

Conflict resolution. Each worker informs their employer of their assignment R_i , and their employer then updates their lists E_i accordingly. In some occasions this may create a conflict, *e.g.*, if Bob was assigned to Cohort 1, *i.e.* R_1 , but his boss, Alice, needs him to fill a crucial spot in her employee list E_2 . Bob will then have to inform his local authority that he wishes to transition to R_2 . This can be done via a dedicated call-center or online.

Bob may also have a personal conflict, for example, he wishes to transition to Cohort 2 in order to tend to a family member in that Cohort. Such conflicts too, will be resolved via referral to the call-center/online form.

The main point is that the authority must only *publish* its address based partition R_1 and R_2 . This will suffice for the majority of citizens, as most will be indifferent as to which week they are active and which they are not. In cases where conflicts do arise – it will be the responsibility of the employer/individual to request a transition. *No central coordination required.*

Treating transition requests. We wish to make AQ as simple and smooth as possible: for the authority - avoiding complex coordination and book-keeping; for the citizens – avoiding the stress of being assigned an inconvenient schedule; and for the employers – allowing a smooth operation, as congruent as possible with business needs (*i.e.* E_1, E_2). Therefore, the conflict resolution policy we recommend is extremely simple and accommodating:

All transition requests from R_1 to R_2 (or vice versa) are, by default, granted.

This saves on bureaucracy and tedious (and futile) efforts for centralized coordination. Hence, in principle, citizens do not *ask* for being transitioned, rather they *inform* the authority that they have transitioned, such that it can update its lists accordingly.

Conditions apply. There are two crucial conditions that apply to the above lenient policy. (i) *Touch-move rule.* Each civilian can only transition once, and prior to a preset deadline T . Hence, once the AQ routine is already running, they remain bound to their assigned cohort. No going back. (ii) *Household is a unit.* Transition requests must preserve the household integrity, namely if Bob wishes to transition from R_1 to R_2 , his cohabitants, Alice and the kids, transition together with him.

How can we be so accommodating? The reason is simple – *we do not need a perfectly balanced partition of the population.* Indeed, having one cohort slightly larger than the other has no bearing on the effectiveness of AQ. In fact, even in our simulations, due to stochastic effects and imperfect balance of household sizes, our partition was never *exactly* 50:50. We can therefore afford to correct the authority's lists R_i to be as compatible as possible with those of the employers E_i , and the individual preferences of individual residents, as, indeed, deviations from an exact partition pose no problem. Statistically, at a national/regional level, the resulting partition will likely be close to 50:50, anyhow.

The outcome. Lacking any coordination at the national level, by the deadline T society will naturally be divided into two cohorts, in which almost all employees/employers are granted their ideal work schedule. *The local authorities need not micro-manage the partitioning, just track it.* Starting with their initial address-based R_{i0} , following all the transition requests, they will eventually end up with the updated lists R_{if} , which they will use to inform schools and other local services.

The only coordination that is required is between household members. In simple words, Bob, if he so wishes, will have to transition together with his cohabitant Alice. And in case of conflict, they will have to resolve it between them. *As always, when living together – compromise is key...*

We have now added a more detailed Implementation section, including a dedicated Box in the paper to (briefly) outline this partitioning policy. We also discuss other challenges, such as enhancing cooperation and coping with defection.

Comment 3

The authors assume that asymptomatic individuals are not very infectious; however, studies have shown that asymptomatic individuals can be major drivers of disease spread and transmission. I am interested in seeing how the results would change if the relative infectiousness of asymptomatic individuals was higher than assumed.

Response

We seem to have been misunderstood here. While we mentioned studies indicating that asymptomatic individuals are less infectious, we are aware that this is debatable, and therefore *at no point in our analysis did we rely on such assumptions*. As we stated in our original text, and now reiterate more clearly

... to err on the side of safety, in our modeling of the spread we use a uniform infection rate, for all individuals - symptomatic or asymptomatic...

Hence, we agree with the Referee that this point is interesting and, potentially, crucial to the effectiveness of AQ. And precisely for this reason, in our modeling, we take the asymptomatic individuals to be *just as infectious* as their symptomatic peers.

Comment 4

Model parameters should be provided in a Table in the Supplementary including relevant citations and brief descriptions. I noticed that the authors provide the parameters in their online code files (on GitHub), but without a description, one gets lost in how the parameters are used. Providing a table will help with reproducibility.

Response

Done. Thanks for pointing this out. In the now elaborated Supplementary Information we include several Tables (1 – 5), summarizing all our model parameters, their meaning and assigned values.

Comment 5

I recommend this article for acceptance, provided the comments here are addressed.

Response

Thanks. We hope that following our revisions, both improving the model and expanding on its practical implementation, the Referee will now find our work suitable for publication in *Nature Communications*.

Reviewers' Comments:

Reviewer #1:

Remarks to the Author:

The authors did a major overhaul of their work. The modeling framework is now adequate for the evaluation of the proposed strategies and thus the manuscript is remarkably improved. I have now only a short list of minor comments.

Original comment: "I understand the difficulty of defining R_0 for such a model, but it is hard to compare results to other studies and even between the baseline scenario and the mitigated ones without knowing R_0 of the system.". My suggestion is to provide at least a rough estimate of R_0 by using the well-known relation between the growth rate of an epidemic and the reproduction number (see for instance <https://royalsocietypublishing.org/doi/abs/10.1098/rspb.2006.3754>): $R_0 \sim 1 + a * b$, where a is the growth rate of the epidemic and b is the length of the generation time. (Of course the authors can use the full equation instead of the simple approximation for the SIR model that I have just written). Given that a was estimated to be 0.26, the authors will probably end up finding R_0 around 2.2-2.8, which would be in line with the literature.

Exactly as for R_0 , which is very useful to provide a context for the performed analyses, it would be important to show the household secondary attack rate in the different scenarios about the transmissibility. This is easy to calculate in the simulations as $\langle (\text{number of infection in household} - 1) / (\text{household size} - 1) \rangle$. To do so, the authors do not need to re-run all simulations, but only to have a look at this key epidemiological indicator in the different transmission scenarios in the absence of interventions. Indeed, the household secondary attack rate was estimated in several studies on COVID-19 epidemiology to be roughly in the range 20%-50%. If the authors find similar estimates in their scenarios, that would strengthen their analysis.

Beta (the growth rate of the epidemic according to the notation used in this manuscript) is obtained by analyzing the time series of new cases by date of reporting. This is well known to represent an overestimation of the actual epidemic growth. This is not a problem here as the authors provide a scenario analysis (i.e., they are not trying to reproduce a specific epidemic trajectory). Still, the fact that this estimate is an overestimation should be acknowledged so that it results clear also to a non-expert reader.

Second to last line of the abstract: "viral spread" -> "infection spread".

Reviewer #2:

None

Reviewer #3:

Remarks to the Author:

Dear Authors,

The authors' revisions have satisfied my concerns about the validity of the manuscript results. I have looked over the main results as well as their comments to me and other reviewers. They have made substantial changes to their model, now incorporating household transmission as well as stochasticity, thus bringing in a new sense of realism.

It still seems that a new social infrastructure would be needed to incorporate the strategies discussed in the manuscript. I am interested in learning about regions and countries where AQ is already implemented. Are these regions as large as the USA or Canada? My concern is that the sharp political divisiveness in the USA will prevent these strategies from ever being implemented.

Nonetheless, the science and the modelling is strong.

Reviewer #1

1. Comment

The authors did a major overhaul of their work. The modeling framework is now adequate for the evaluation of the proposed strategies and thus the manuscript is remarkably improved. I have now only a short list of minor comments.

Response

We wish to thank the Referee for prompting us significantly improve our paper – both technically and in terms of its presentation and context. We are now, indeed, much more confident in our results, and in the relevance of our modeling framework.

2. Comment

*Original comment: "I understand the difficulty of defining R_0 for such a model, but it is hard to compare results to other studies and even between the baseline scenario and the mitigated ones without knowing R_0 of the system.". My suggestion is to provide at least a rough estimate of R_0 by using the well-known relation between the growth rate of an epidemic and the reproduction number (see for instance <https://royalsocietypublishing.org/doi/abs/10.1098/rspb.2006.3754>): $R_0 \sim 1 + a * b$, where a is the growth rate of the epidemic and b is the length of the generation time. (Of course the authors can use the full equation instead of the simple approximation for the SIR model that I have just written). Given that a was estimated to be 0.26, the authors will probably end up finding R_0 around 2.2-2.8, which would be in line with the literature.*

Response

This is very helpful, as we were, indeed struggling to evaluate R_0 for the elaborate and complex COVID-19 disease cycle. We have now calculated it using the Referee's suggestion (Supplementary Note 1.5). Taking a to be the growth rate (β in our notation) and evaluating b from the mean duration of an individual's infectious phase. We arrive at $R_0 \approx 2.4$, which is, encouragingly, within the range observed in the literature.

3. Comment

Exactly as for R_0 , which is very useful to provide a context for the performed analyses, it would be important to show the household secondary attack rate in the different scenarios about the transmissibility. This is easy to calculate in the simulations as $\langle (\text{number of infection in household} - 1) / (\text{household size} - 1) \rangle$. To do so, the authors do not need to re-run all simulations, but only to have a look at this key epidemiological indicator in the different transmission scenarios in the absence of interventions. Indeed, the household secondary attack rate was estimated in several studies on COVID-19 epidemiology to be roughly in the range 20%-50%. If the authors find similar estimates in their scenarios, that would strengthen their analysis.

Response

The Referee distinguishes between our parameter α , which quantifies the overall contribution of in-house transmission to the spread, and the commonly measured *in-house infection rate* ρ , which denotes the probability of an individual to be infected by one of his/her cohabitants. We agree, the latter can be directly extracted from our existing data, and we now, following this comment, do precisely that, and report ρ 's observed value in the relevant location. We find that across the different scenarios, in which our α ranges from $\alpha \sim 0.15$ to $\alpha \sim 0.3$, ρ is between 20 – 40%, congruent with epidemiological analyses.

We wish to emphasize that in the present context our parameter α , even if not *the* most common measure for in-house infections, is a most relevant observable to evaluate quarantine efficiency. Indeed, the infection rate ρ measures the risk for household members to be infected by their cohabitant, but provides no direct insight on the role that such in-house transmission plays in the total spread. For example, even if $\rho = 100\%$, in house transmission may be marginal if, *e.g.*, households are very small, and hence few infections occur at home, or if external infections are, in comparison, much more prevalent. In contrast, α accounts not just for the household attack rate, but also for the distribution of household sizes, and their relative role as compared to the external transmission. This is highly relevant as quarantine plays precisely on that tradeoff – reducing external transmission at the price of exacerbating in-house interactions.

4. Comment

Beta (the growth rate of the epidemic according to the notation used in this manuscript) is obtained by analyzing the time series of new cases by date of reporting. This is well known to represent an overestimation of the actual epidemic growth. This is not a problem here as the authors provide a scenario analysis (i.e., they are not trying to reproduce a specific epidemic trajectory). Still, the fact that this estimate is an overestimation should be acknowledged so that it results clear also to a non-expert reader.

Response

Indeed. We now added this clarification in the relevant location in the paper.

Comment

Second to last line of the abstract: “viral spread” -> “infection spread”.

Response

Thanks. This comment is no longer relevant as we have now reworded our abstract to fit into the 150 word limit.

Reviewer #3

Comment

Dear Authors,

The authors' revisions have satisfied my concerns about the validity of the manuscript results. I have looked over the main results as well as their comments to me and other reviewers. They have made substantial changes to their model, now incorporating household transmission as well as stochasticity, thus bringing in a new sense of realism.

Response

Thanks for helping us improve our work, and especially for prompting us to think more deeply on the implementation at scale.

Comment

It still seems that a new social infrastructure would be needed to incorporate the strategies discussed in the manuscript. I am interested in learning about regions and countries where AQ is already implemented. Are these regions as large as the USA or Canada? My concern is that the sharp political divisiveness in the USA will prevent these strategies from ever being implemented. Nonetheless, the science and the modelling is strong.

Response

We agree. But of course, no such infrastructures and policies will be established unless we communicate this idea to as broad an audience as possible. Unfortunately, we also agree with the Referee that, at present, the US seems to be too divided for such an ambitious, nation-wide strategy. Other countries, such as France, Israel or Germany, as well as several South American countries that have shown significant interest, are more likely to follow this path. Some, like Germany or Israel have already implemented AQ in limited scale (schooling, government branches, large corporations), and we hope that a visible venue such as *Nature Communications* will help us further advance the cause.